# *ATP6V0A1* encoding the a1-subunit of the V0 domain of vacuolar H+-ATPases is essential for brain development in humans and mice

Kazushi Aoto [1✉], Mitsuhiro Kato [2], Tenpei Akita [3], Mitsuko Nakashima[1], Hiroki Mutoh[1], Noriyuki Akasaka[4,11], Jun Tohyama[4], Yoshiko Nomura[5,12], Kyoko Hoshino[5,13], Yasuhiko Ago [6,14], Ryuta Tanaka[7], Orna Epstein[8], Revital Ben-Haim[8], Eli Heyman[8], Takehiro Miyazaki [1], Hazrat Belal [1], Shuji Takabayashi[9], Chihiro Ohba[10], Atsushi Takata[10], Takeshi Mizuguchi [10], Satoko Miyatake[10], Noriko Miyake[10], Atsuo Fukuda [3], Naomichi Matsumoto [10✉] & Hirotomo Saitsu [1✉]

Vacuolar H+-ATPases (V-ATPases) transport protons across cellular membranes to acidify various organelles. *ATP6V0A1* encodes the a1-subunit of the V0 domain of V-ATPases, which is strongly expressed in neurons. However, its role in brain development is unknown. Here we report four individuals with developmental and epileptic encephalopathy with *ATP6V0A1* variants: two individuals with a de novo missense variant (R741Q) and the other two individuals with biallelic variants comprising one almost complete loss-of-function variant and one missense variant (A512P and N534D). Lysosomal acidification is significantly impaired in cell lines expressing three missense ATP6V0A1 mutants. Homozygous mutant mice harboring human R741Q (*Atp6v0a1R741Q*) and A512P (*Atp6v0a1A512P*) variants show embryonic lethality and early postnatal mortality, respectively, suggesting that R741Q affects V-ATPase function more severely. Lysosomal dysfunction resulting in cell death, accumulated autophagosomes and lysosomes, reduced mTORC1 signaling and synaptic connectivity, and lowered neurotransmitter contents of synaptic vesicles are observed in the brains of *Atp6v0a1A512P/A512P* mice. These findings demonstrate the essential roles of *ATP6V0A1/Atp6v0a1* in neuronal development in terms of integrity and connectivity of neurons in both humans and mice.

[1] Department of Biochemistry, Hamamatsu University School of Medicine, Hamamatsu, Japan. [2] Department of Pediatrics, Showa University School of Medicine, Tokyo, Japan. [3] Department of Neurophysiology, Hamamatsu University School of Medicine, Hamamatsu, Japan. [4] Department of Child Neurology, National Hospital Organization Nishiniigata Chuo Hospital, Niigata, Japan. [5] Segawa Neurological Clinic for Children, Tokyo, Japan. [6] Department of Neonatology, Ibaraki Children's Hospital, Mito, Japan. [7] Ibaraki Pediatric Education and Training Station, University of Tsukuba, Mito, Japan. [8] Pediatric Neurology and Development Center, Shamir Medical Center, Tzrifin, Beer Yaakov, Israel. [9] Laboratory Animal Facilities & Services, Preeminent Medical Photonics Education & Research Center, Hamamatsu University School of Medicine, Hamamatsu, Japan. [10] Department of Human Genetics, Yokohama City University Graduate School of Medicine, Yokohama, Japan. [11]Present address: Department of Pediatrics, Niigata Prefecture Hamagumi Medical Rehabilitation Center for Disabled Children, Niigata, Japan. [12]Present address: Yoshiko Nomura Neurological Clinic for Children, Tokyo, Japan. [13]Present address: Segawa Memorial Neurological Clinic for Children, Tokyo, Japan. [14]Present address: Department of Pediatrics, Graduate School of Medicine, Gifu University, Gifu, Japan. ✉email: kaz@hama-med.ac.jp; naomat@yokohama-cu.ac.jp; hsaitsu@hama-med.ac.jp

The vacuolar $H^+$-ATPase (V-ATPase) is an ATP-dependent proton pump that transports protons across cellular membranes to acidify various organelles, such as lysosomes, endosomes, the trans-Golgi network, and secretory granules, including synaptic vesicles. The acidification of these organelles is required for many cellular processes, including protein degradation, receptor-mediated endocytosis, and proton-coupled transport of small molecules[1–3]. In addition, recent studies suggest that the presynaptic V-ATPase modulates neurotransmitter exocytotic release[1,4]. V-ATPase also plays important roles in a variety of cellular signaling pathways, including the mechanistic target of rapamycin (mTor) complex 1 (mTORC1), AMP-activated protein kinase, Wnt, and Notch signaling[3]. The V-ATPase is a large multisubunit complex, composed of an ATP hydrolytic domain (V1) and a proton translocation domain (V0) that operates by a rotary mechanism. Consistent with the important roles of V-ATPase in cellular processes, variants in seven genes encoding some subunits of the V0 or V1 domain are associated with human diseases. Biallelic variants in six genes are related to renal tubular acidosis (ATP6V1B1 and ATP6V0A4), osteopetrosis with neurodegeneration (ATP6V0A3), and cutis laxa (ATP6V0A2, ATP6V1E1, and ATP6V1A), and monoallelic heterozygous variants in ATP6V1B2 are associated with deafness–onychodystrophy syndrome and Zimmermann–Laband syndrome type 2, which shows facial dysmorphisms and intellectual disability[5,6]. In addition, de novo variants in ATP6V1A have been recently shown in individuals with developmental and epileptic encephalopathy (DEE)[7]. These suggest that various degrees of V-ATPase impairment cause disorders with a wide phenotypic spectrum. Lysosomal impairment is a common feature of these disorders[5–7]. Proton translocation occurs through the integral V0 domain, which is composed of single copies of the a, d, and e-subunits, and a hexameric ring of very hydrophobic subunits (c and c″-subunits) in vertebrates[1–3]. Four different isoforms of the a-subunit (a1–a4) encoded by different genes have been identified in humans and mice. These isoforms are expressed in a tissue-specific manner, and the a1 isoform is strongly expressed in neurons[8].

Here, we identify de novo and biallelic variants in ATP6V0A1, encoding the a1-subunit of the V0 domain, in individuals with DEE who showed various degrees of intellectual disability, developmental delay, and epilepsy. We demonstrate impairment of lysosomal acidification in cells stably expressing each of the three missense mutants. In addition, lysosomal abnormalities, cell death, accumulated autophagosomes and lysosomes, downregulation of mTORC1 signaling, and reduced synaptic connectivity and neurotransmitter contents of synaptic vesicles are observed in mice harboring a homozygous missense variant identified in humans. Our data highlight the essential roles of the a1-subunit of the V0 domain of V-ATPase for brain development in terms of integrity and connectivity of neurons in both humans and mice.

## Results

**Identification of de novo and biallelic ATP6V0A1 variants.** We performed whole-exome sequencing (WES) for a total of 700 individuals with DEE. Trio-based WES of 211 families with DEE revealed a de novo ATP6V0A1 variant (NM_001130020.1: c.2222G>A, p.(Arg741Gln)) in individual 1 (Fig. 1a). By searching the remaining 489 case-only WES data for possible pathogenic ATP6V0A1 variants, we found the identical c.2222G>A variant in individual 2 though we were unable to confirm de novo occurrence because his father was deceased. In addition, we searched the denovo-db database[9] and found two additional cases of the de novo c.2222G>A variant in the Deciphering Developmental Disorders project[10], suggesting that the recurrent de novo

c.2222G>A variant is likely to be associated with neurodevelopmental phenotypes. Furthermore, we identified two individuals with biallelic ATP6V0A1 variants: individual 3 has a missense variant (c.1534G>C, p.(Ala512Pro)) and a 50-kb deletion involving ATP6V0A1 (del(17)(q21.2)), which were transmitted from his father and mother, respectively, and individual 4 has a splice site variant (c.196+1G>A) and a missense variant (c.1600A>G, p.(Asn534Asp)), which were transmitted from his father and mother, respectively (Fig. 1 and Supplementary Fig. 1). The c.2222G>A and c.1534G>C variants are absent in the Genome Aggregation Database (gnomAD)[11]. The c.196+1G>A and c.1600A>G variants are found in 2 of 251,356 alleles and 2 of 251,286 alleles, respectively, in the gnomAD database, indicating that these variants are very rare. In silico prediction tools suggest that the three missense variants could affect function and c.196+1G>A would disrupt the splice donor site (Supplementary Table 1). ATP6V0A1 showed a high Z score (3.74) for missense variants and a high pLI score (1.0) for loss-of-function variants, suggesting that ATP6V0A1 is intolerant to variant. While Ala512 and Asn534 are located at the vacuolar side, and conserved among mammals and vertebrates, Arg741 is located at the transmembrane domain and highly conserved from humans to yeast (Fig. 1b). In fact, Arg735 of the yeast V-ATPase VPH1 and Arg755 of the drosophila VHA100-1, which is homologous to Arg741 of the human ATP6V0A1, has been shown to be essential for proton transport[12,13], strongly suggesting that the de novo R741Q variant in individuals 1 and 2 affects the transport function of V-ATPase. On the other hand, individuals 3 and 4 have two variant alleles comprising one likely loss-of-function allele (a 50-kb deletion involving exons 1–13 of ATP6V0A1 (Fig. 1c, d) or a splice site variant) and one hypomorphic allele (A512P or N534D). Therefore, it is postulated that various degrees of impairment of ATP6V0A1 function with both dominant and recessive inheritance are associated with neurodevelopmental disorders.

**Clinical features of individuals with ATP6V0A1 variants.** Clinical features of four individuals with ATP6V0A1 variants are summarized in Supplementary Table 2, and case reports of three individuals are described in the Supplementary Text. Clinical information for the two individuals with the c.2222G>A variant from the Deciphering Developmental Disorders project is unavailable. The initial symptom of the two individuals with the de novo c.2222G>A variant (individuals 1 and 2) was developmental delay at 5 months of age, and they developed seizures soon after. Seizure onset of the two individuals with biallelic ATP6V0A1 variants (individuals 3 and 4) was at ~1 month of age. Seizures were intractable in all four individuals and intellectual disability was profound in three individuals (individuals 1, 2, and 3). Three individuals (individuals 1, 2, and 4) were ambulant. Electroencephalogram showed focal epileptic discharges, hypsarrhythmia, and suppression burst pattern in the individuals 1, 2 and 3, respectively (Supplementary Fig. 2). Brain MRI showed enlarged lateral ventricles in patient 1, and severe brain atrophy including cerebellum in individuals 3 and 4 (Fig. 2), suggesting that the cerebellum was commonly affected in the two individuals with biallelic ATP6V0A1 variants.

**Lysosomal acidification is impaired in cells expressing ATP6V0A1 variants.** To examine the effects of ATP6V0A1 variants on the cellular localization and lysosomal acidification, we established stable human embryonic kidney cell lines (HEK293FT) expressing C-terminal 3×HA-tagged human ATP6V0A1 (ATP6V0A1-3×HA) and stable neuroblastoma-2A (N2A) cell lines expressing N-terminal HA-tagged human

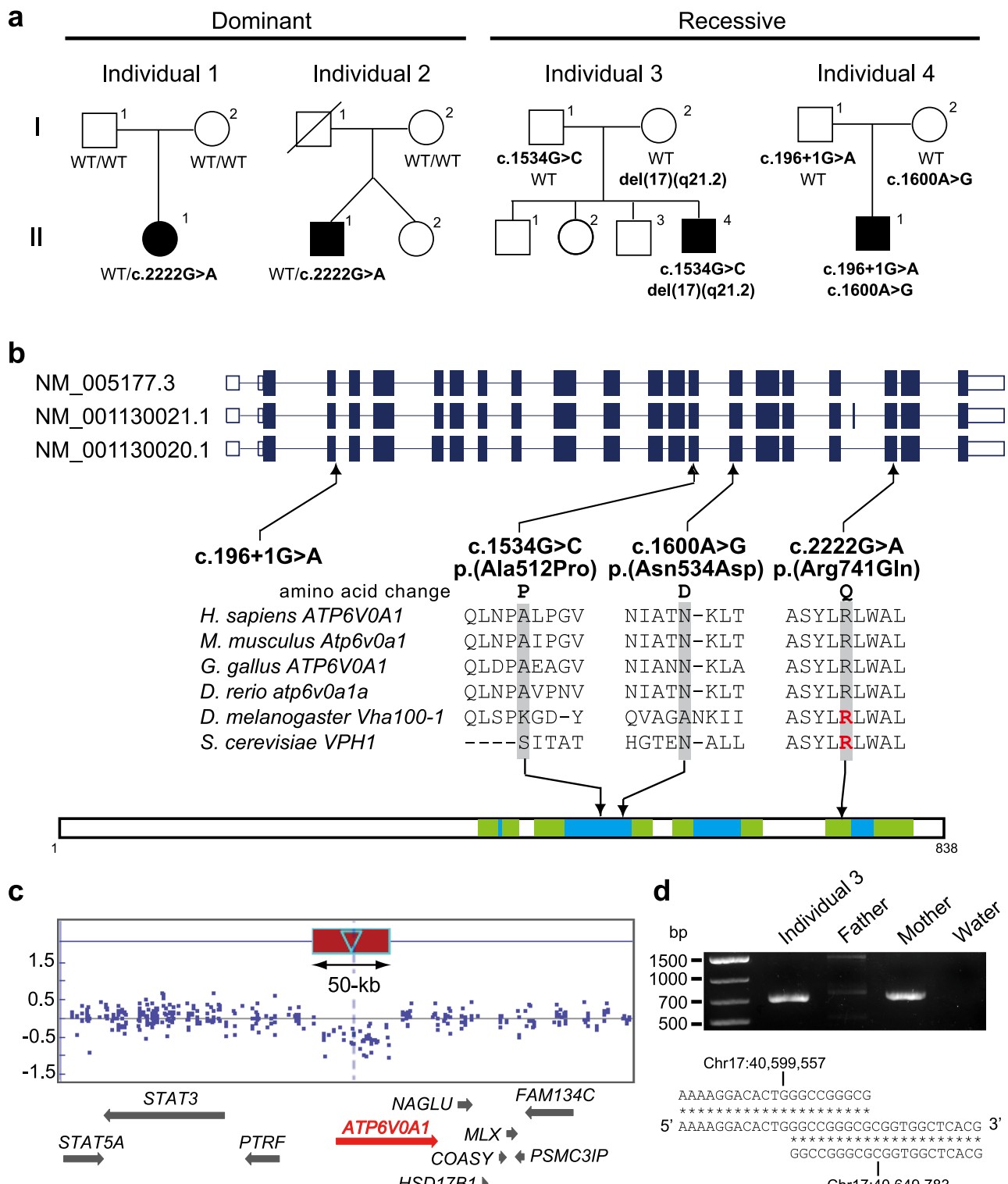

**Fig. 1 De novo and biallelic variants in *ATP6V0A1* in individuals with DEE. a** Familial pedigrees of four individuals with *ATP6V0A1* variants. The segregation of each variant is shown. **b** *ATP6V0A1* transcripts (UTR and coding region are open and filled rectangles, respectively). Three transcripts are registered in the RefSeq database, and all four variants are involved in common exons of the three transcripts. Human ATP6V0A1 is 838 aa in length (NP_001123492.1 from NM_001130020.1) and contains nine putative transmembrane domains according to UniProt (Q93050). Amino acids located at the cytoplasmic and vacuolar sides and in the transmembrane domain are shown in white, sky blue, and green, respectively. The multiple sequence alignment was performed via the CLUSTALW website (https://www.genome.jp/tools-bin/clustalw) using the following sequences: NP_001123492.1 (*Homo sapiens*), NP_058616.1 (*Mus musculus*), NP_990055.1 (*Gallus gallus*), NP_997837.1 (*Danio rerio*), NP_651672.1 (*Drosophila melanogaster*), and NP_014913.3 (*Saccharomyces cerevisiae*). Arg735 in VPH1 and Arg755 in VHA100-1, which are essential for proton transport, are highlighted in red. **c** A 50-kb deletion involving *ATP6V0A1* in individual 3 detected by cytogenetics whole-genome 2.7 M array. The deleted region is indicated by a red box. **d** Breakpoint-specific PCR and sequencing confirmed the 50-kb deletion (chr17: 40,599,557–40,649,783), which was transmitted from his mother. Top, middle, and bottom strands show proximal, recombined, and distal sequences, respectively. A total of 9-bp sequences were overlapped in the proximal and distal breakpoints. Data in **d** are representative of two independent experiments.

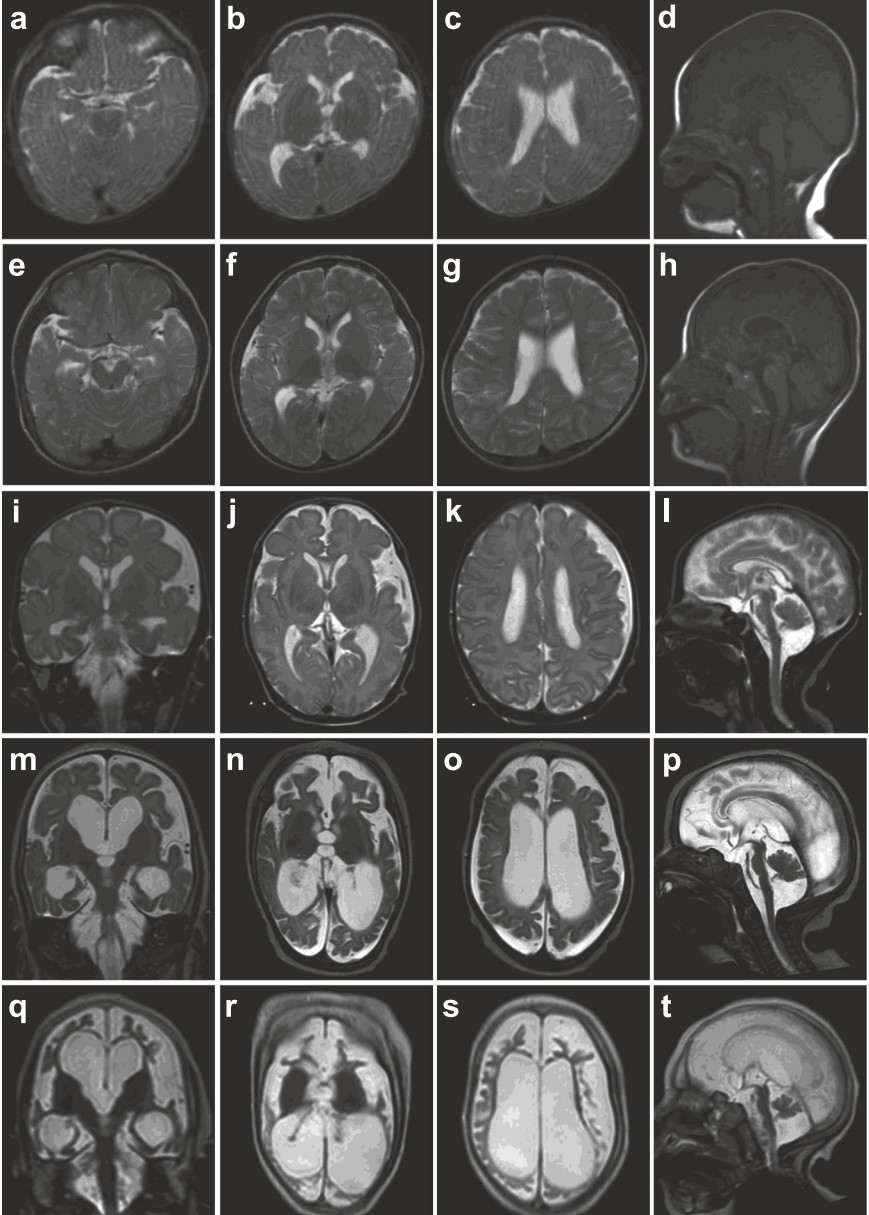

**Fig. 2 Brain MRI of individuals with the *ATP6V0A1* variant.** Individual 2 at 6 months (**a–d**) and at 4 years of age (**e–h**), and individual 3 at 10 days (**i–l**), 6 months (**m–p**), and 3 years of age (**q–t**). All images are T2-weighted except for **d** and **h**, which are T1-weighted. Axial images of individual 2 (**a–c**, **e–g**) show mildly enlarged lateral ventricles, which did not change in size between 6 months and 4 years. Sagittal images of individual 2 (**d** and **h**) show a mildly enlarged third ventricle, which was not progressive, and the cerebellum and brainstem are normal. Individual 3 shows progressive atrophy of the cerebrum, including the hippocampi and cerebellum associated with enlarged sulci, ventricles, and subarachnoidal space. The cerebellum of individual 3 showed hypoplasia at 10 days (**l**). Basal ganglia and brainstem except the pontine basis are relatively spared.

ATP6V0A1 (HA-ATP6V0A1). Co-staining with antibodies for HA-tag, ATP6V1A (A-subunit of the V1 domain), and lysosomal marker Lamp2 revealed that wild-type ATP6V0A1 (ATP6-V0A1^WT-3×HA) was partially colocalized with ATP6V1A and Lamp2, which suggests that assembled V-ATPase complex, including ATP6V0A1 and ATPV1A was localized in the lysosome (Fig. 3a). The localizations of three ATP6V0A1 mutants (ATP6V0A1^A512P-3×HA, ATP6V0A1^N534D-3×HA, and ATP6-V0A1^R741Q-3×HA) were similar to those of wild-type ATP6V0A1 (Fig. 3a). However, the pH measurement of HEK293FT cell lines using LysoSensor Yellow/Blue dextran revealed that lysosomal pH values of all the three types of mutant expressing HEK293FT cells were significantly higher than those of wild-type expressing cells (Fig. 3b). The impaired acidification was also

confirmed in N2A stable cell lines using LysoTracker. While LysoTracker produced strong red fluorescent signals in lysosomes of untransfected control and wild-type ATP6V0A1 expressing N2A cells, the dye produced significantly weaker fluorescent signals in all the mutant expressing cells (Fig. 3c, d). These data suggested that all ATP6V0A1 missense variants impaired lyso-somal acidification in cell lines.

**Mortality of three *Atp6v0a1* mutant mice suggests degree of loss-of-function of R741Q and A512P variants**. To further explore the pathophysiology of DEE caused by *ATP6V0A1* variants, we generated mutant mice models harboring a human A512P variant (A506P in mice; NCBI reference number NP_058616.1) and a human R741Q variant (R741Q in mice),

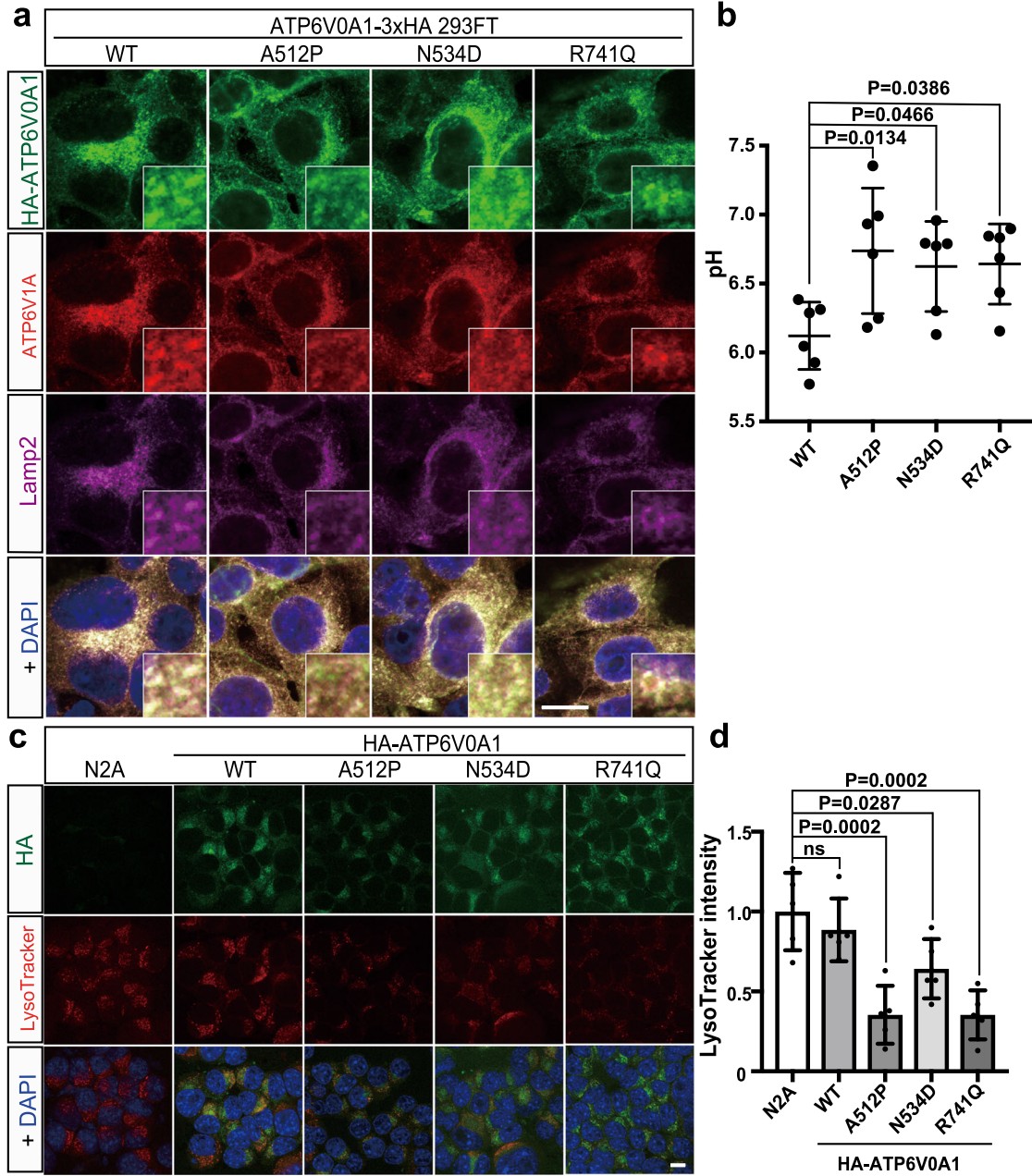

**Fig. 3 Cellular localization and abnormal lysosomal acidification of ATP6V01A1 mutants. a** Stable HEK293FT cell lines expressing ATP6V0A1-3×HA were stained with antibodies against HA (green), ATP6V1A (red), and a lysosomal marker, Lamp2 (purple), showing partial colocalization of ATP6V0A1, ATP6V1A, and Lamp2. Magnified inserts showed colocalization of HA-ATP6V0A1, ATP6V1A, and Lamp2. Scale bars, 10 μm. **b** Quantification of lysosomal pH in the cell lines expressing wild-type (WT) and three different mutants of ATP6V0A1-3×HA using LysoSensor. All three ATP6V0A1 mutants caused elevated pH compared with wild-type ATP6V0A1. Data represented as mean values ± standard deviation of six independent calculations. **c** Wild-type and mutant HA-ATP6V0A1 expressing stable neuro2a cell lines were stained with an anti-HA antibody (green) and LysoTracker fluorescence dye (red). Scale bars, 10 μm. **d** Quantification of the LysoTracker intensity of untransfected neuro2a cells, HA-ATP6V0A1 wild-type, and mutant expressing stable cell lines. The intensity was significantly reduced in all the mutant expressing cells compared with untransfected control. Data represented as mean values ± standard deviation of five measurements. Ordinary one-way ANOVA with Dunnett's multiple comparison test was used for comparing among ATP6V0A1 mutants (**b**, **d**). Cell image data in **a** and **c** are representative of more than three independent experiments. Source data are provided as a Source data file.

using the CRISPR-Cas9 genome editing and electroporation method (Supplementary Fig. 3)[14]. To clearly indicate corresponding human variants, we designate two mice lines as *Atp6v0a1*[A512P] and *Atp6v0a1*[R741Q] in this report. During the generation of *Atp6v0a1*[A512P] mice, we incidentally obtained a mutant with 1-bp deletion that caused a frameshift (Supplementary Fig. 3f), and we designate this mouse line as *Atp6v0a1* knockout (*Atp6v0a1*[KO]). Heterozygous mice of three lines

(*Atp6v0a1*[A512P/+], *Atp6v0a1*[R741Q/+], and *Atp6v0a1*[KO/+]) showed no obvious abnormalities over 6 months of age and were fertile. However, no homozygous pups of two mice lines (*Atp6v0a1*[R741Q/R741Q] and *Atp6v0a1*[KO/KO]) were born. The embryonic development of *Atp6v0a1*[KO/KO] and *Atp6v0a1*[R741Q/R741Q] mice was stopped at embryonic days 5–6 (E5–6) before gastrulation (Supplementary Fig. 4). Thus, the R741Q variant may lead to the almost complete loss-of-function of Atp6v0a1

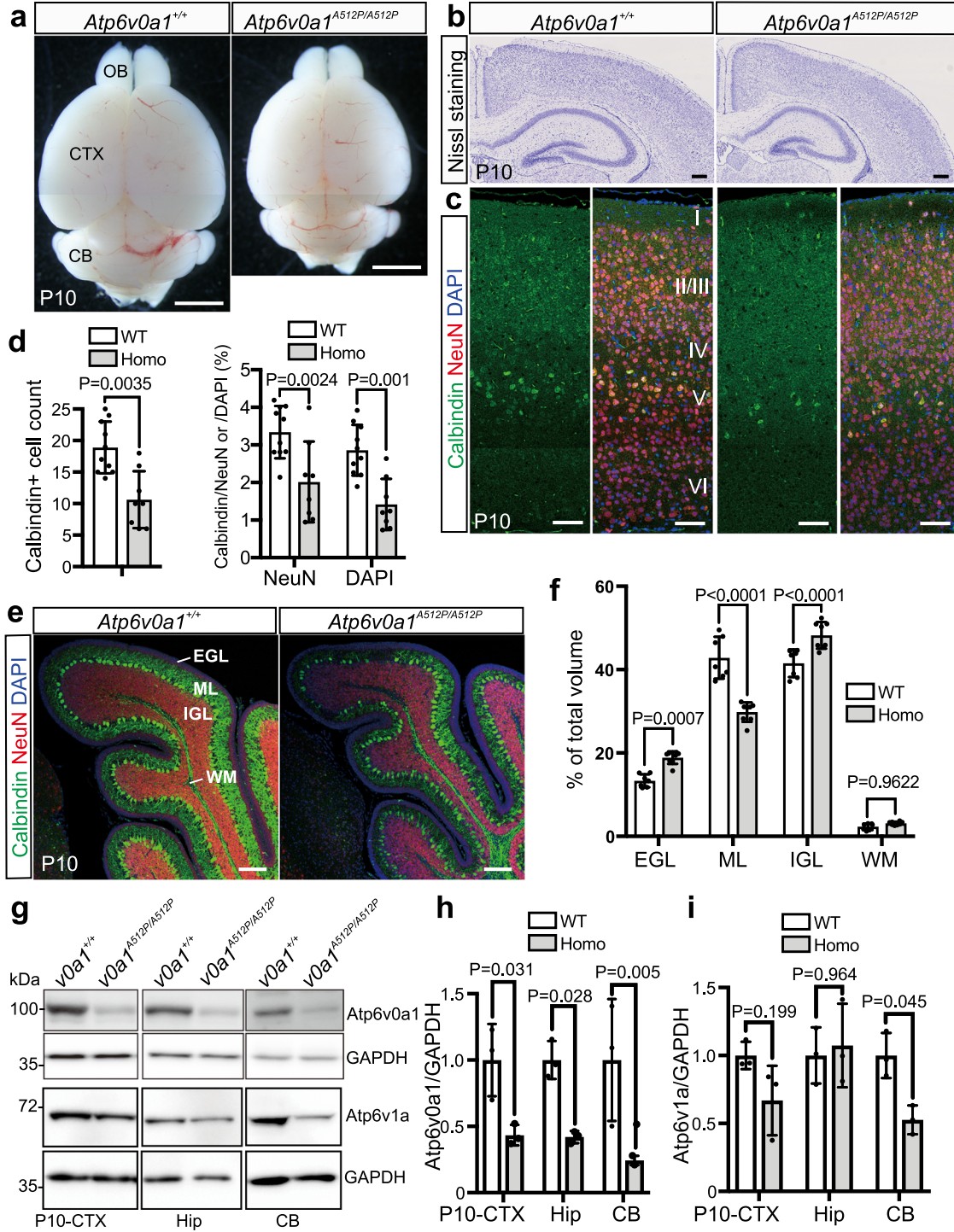

comparable to the frameshift variant. Although we did obtain homozygous *Atp6v0a1^A512P/A512P* pups, their body weights were lower than that of wild-type pups and they died at ~2 weeks of age (Supplementary Fig. 4). These data suggest that both A512P and R741Q variants cause loss-of-function of Atp6v0a1, and R741Q affects V-ATPase function more severely. One of the causes of death in *Atp6v0a1^A512P/A512P* pups was a cleft palate that prevented the intake of their mother's milk, which was recognized in 7% of homozygous pups (Supplementary Fig. 5).

**Impaired neural development and synapse formation in the *Atp6v0a1^A512P/A512P* mutant brain.** Notably, impaired

motor function was observed at postnatal day 3 (P3) of *Atp6v0a1^A512P/A512P* pups (Supplementary Fig. 4f) and long-surviving *Atp6v0a1^A512P/A512P* pups showed ataxia (Supplemental Movie), which seems to be consistent with the human phenotype. The brain size of *Atp6v0a1^A512P/A512P* pups was smaller than that of wild-type pups at P10 (Fig. 4a), although the size at E18.5 and P1 was normal (data not shown). Gross brain structural abnormalities were not obvious in Nissl-stained coronal sections (Fig. 4b); however, double staining of the cortical layer V pyramidal neuron marker calbindin and the neural marker NeuN indicated the reduced numbers of layer V neurons in *Atp6v0a1^A512P/A512P* pups at P10 (Fig. 4c, d). The

**Fig. 4 Abnormal cortical and cerebellar layers and decreased Atp6v0a1 protein expression in *Atp6v0a1^A512P/A512P* pups. a** The dorsal view of *Atp6v0a1^+/+* (WT) and *Atp6v0a1^A512P/A512P* (Homo) brains at postnatal 10 day (P10), showing the smaller size of the *Atp6v0a1^A512P/A512P* brain. Scale bars, 2 mm. **b** Nissl-stained coronal sections of the cortex (CTX) and hippocampus (Hip) in *Atp6v0a1^+/+* and *Atp6v0a1^A512P/A512P* pups at P10 showed no gross structural abnormalities. Scale bars, 200 μm. **c** Immunostaining of the cortical layer V marker calbindin (green) and the neuronal marker NeuN (red) in coronal sections of *Atp6v0a1^+/+* and *Atp6v0a1^A512P/A512P* cortexes at P10. Nuclei were stained with DAPI (blue). Scale bars, 50 μm. **d** The number of calbindin-positive cells (left chart) and the ratios of calbindin-positive/NeuN-positive cells and of calbindin-positive/DAPI-positive cells within a 200-μm-wide bin (right chart) were decreased in the cortical layer of *Atp6v0a1^A512P/A512P* pups at P10. Data represented as mean values ± standard deviation of eight or more counts. **e** Sagittal sections of the cerebellums in *Atp6v0a1^+/+* and *Atp6v0a1^A512P/A512P* pups at P10 stained with antibodies against calbindin as a Purkinje cell marker (green) and NeuN (red). Scale bars, 50 μm. **f** Comparison of layer volumes of external granule layer (EGL), molecular layer (ML), internal granule layer (IGL), and white matter (WM). Reduced ML volume was evident, suggesting that dendrite development of Purkinje cells would be impaired. Data represented as mean values ± standard deviation of seven or more measurements. **g** Immunoblot analysis of Atp6v0a1, Atp6v1a, and GAPDH loading control in the CTX, Hip, and cerebellum (CB) of *Atp6v0a1^+/+* and *Atp6v0a1^A512P/A512P* pups at P10. **h, i** Quantification of the band intensity of the immunoblot, which shows reduced levels of Atp6v0a1 protein in all the three regions of *Atp6v0a1^A512P/A512P* brains (**h**) and reduced level of Atp6v1a in cerebellum (**i**). Data represented as mean values ± standard deviation of three independent experiments. Paired *t* test (**d**, left), two way ANOVA with Sidak's multiple comparisons test (**d**, right), or two way ANOVA with Bonferroni's multiple comparisons test (**f, h, i**) was used. Data in **a–c**, **e**, and **g** are representative of more than three independent experiments. Source data are provided as a Source data file.

cerebellums of *Atp6v0a1^A512P/A512P* pups had normal lobules I–X, but showed a reduced molecular layer (ML; Fig. 4e, f), suggesting that dendrite development of Purkinje cells would be impaired. Moreover, to examine the neuronal connection in wild-type and mutant mice, double immunostaining of the presynaptic marker VGLUT1 and the postsynaptic marker PSD95 was performed in hippocampal CA3 regions and cerebellar lobules. Excitatory synapses are provided by mossy fibers in stratums lucidum and radiatum in the hippocampal regions, and by parallel fibers in MLs of the cerebellar lobules. VGLUT1 and PSD95 double-positive excitatory synapses were dramatically reduced in both the hippocampal and cerebellar regions in *Atp6v0a1^A512P/A512P* pups at P10 (Fig. 5). Thus, *ATP6V0A1^A512P/A512P* mice showed defects in the neuronal development and synapse formation.

**Atp6v0a1 protein level is decreased in the *Atp6v0a1^A512P/A512P* mutant brain**. To examine in which cell types *Atp6v0a1* is expressed in the mouse brain, we performed double immunofluorescence staining of Atp6v0a1 and neuronal/glial cell type markers. The results suggested that most Atp6v0a1 proteins were present in NeuN-positive neurons in the cortex and in the dentate gyrus, CA1 and CA3 regions of the hippocampus (Supplementary Fig. 6a), whereas fewer Atp6v0a1 were overlapped with an astrocyte marker GFAP, an oligodendrocyte marker CNPase, and a microglia marker IbaI (Supplementary Fig. 6b). In the cerebellum, Atp6v0a1 proteins were present in Purkinje and granule cells (Supplementary Fig. 6c). The Atp6v0a1 protein levels were significantly decreased in the cortex, hippocampus, and cerebellum of *Atp6v0a1^A512P/A512P* pups at P10 (Fig. 4g, h), and a tendency toward reduced level was observed as early as at P1 (Supplementary Fig. 6d). We also examined protein level of another major V-ATPase subunit, Atp6v1a (A-subunit of the V1 domain), and found that Atp6v1a level was significantly decreased in cerebellum of *Atp6v0a1^A512P/A512P* pups (Fig. 4g, i). These data suggest that *Atp6v0a1^A512P* may affect protein stability of Atp6v0a1, which could affect the protein levels of other V-ATPase subunits in vivo. To test this possibility, we performed in vitro cycloheximide assay using ATP6V0A1-3×HA HEK293FT stable cell lines (Supplementary Fig. 7a, b). ATP6V0A1^WT-3×HA proteins were stable and only 20% reduction was observed after 8 h cycloheximide treatment. In this condition, we found no differences among protein levels of wild-type and three mutant ATP6V0A1-3×HA. These data suggest that the A512P variant could affect protein levels in vivo, but our in vitro test may be not sensitive enough to detect this difference during 8 h periods. ATP6V0A1 proteins undergo glycosylation[15], and glycosylation

difference might contribute to the pathogenicity of three variants. However, the glycosylation of three mutants was not changed compared with that of wild-type HA-ATP6V0A1 (Supplementary Fig. 7c).

**Increased neuronal cell death, abnormal cellular distribution of lysosomes, and decreased cathepsin D activity in *Atp6v0a1^A512P/A512P* mutants**. Lysosomal dysfunction has been shown to induce inactivation of lysosomal protease and caspase-dependent cell death, which is triggered by a release of lysosomal cathepsins to the cytoplasm[16,17]. Therefore, we first performed TdT-mediated dUTP nick-end labeling (TUNEL) analysis to detect neuronal cell death. We found that many TUNEL-positive cells were present in the cortex and hippocampus of *Atp6v0a1^A512P/A512P* pups (Fig. 6a, b). Next, we examined the subcellular distribution of cathepsin D by immunostaining. In wild-type pups, cathepsin D was localized as puncta of lysosomes in the cytoplasm (Fig. 6c). However, in *Atp6v0a1^A512P/A512P* pups, large dense cathepsin D accumulations were noticeable and they were colocalized with abnormally dispersed distributions of lysosome-associated membrane protein 1 (Lamp1) in the hippocampal CA3 region and dentate gyrus, and they were relatively few in cortex at P10 (Fig. 6c and Supplementary Fig. 8). Immunoblotting revealed that the mature form of cathepsin D was significantly reduced in hippocampus and cerebellum (Fig. 6d, arrow, e). These results suggested that lysosomal activity was impaired in *Atp6v0a1^A512P/A512P* pups, which caused abnormal cellular distribution of lysosomes and may be related to neuronal death.

**Autophagosome and lysosome accumulation in *Atp6v0a1^A512P/A512P* mutants**. V-ATPase inhibition in neural cell lines fails to maintain an acid pH and leads to lysosomal dysfunction[15,18]. During macroautophagy, the fusion between autophagosomes and lysosomes is important for autophagosome clearance[19,20]. These reports raise the possibility of the failure of autophagosome clearance and the resulting accumulation of autophagosomes in the neurons of *Atp6v0a1^A512P/A512P* pups. Therefore, we next examined the expression of microtubule-associated protein 1 light chain 3 (LC3), an autophagosome marker. Immunostaining of LC3 demonstrated the accumulation of LC3 puncta in the neurites and cell bodies of the cortical pyramidal cells in layer V of *Atp6v0a1^A512P/A512P* pups (Fig. 7a, arrows and arrowhead). In some pyramidal neurons, LC3 puncta were associated with diffuse distributions of Lamp2, another marker of lysosome (Fig. 7a, arrowheads). Similar findings were

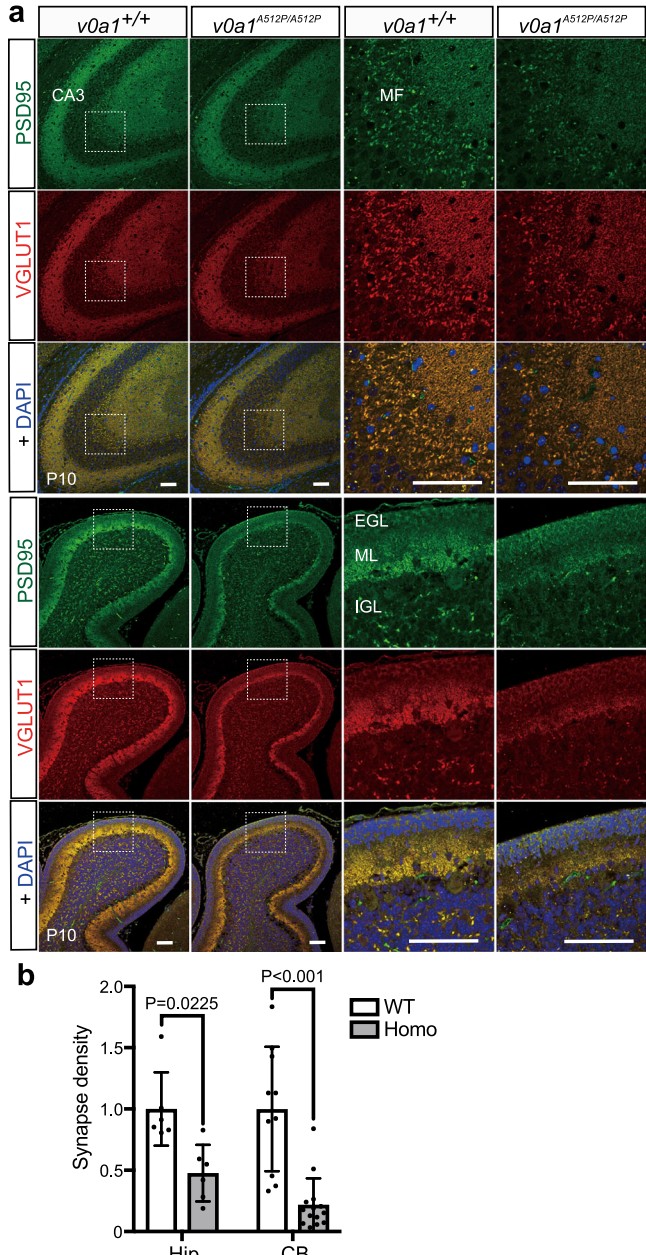

**Fig. 5 Excitatory synaptic densities in hippocampal CA3 and cerebellar molecular layers of *Atp6v0a1*$^{+/+}$ and *Atp6v0a1*$^{A512P/A512P}$ brains.**
**a** Immunostaining of a postsynaptic marker PSD95 (green) and a presynaptic marker VGLUT1 (red) in hippocampal CA3 (upper three rows) and in a cerebellar lobule (lower three rows) obtained from *Atp6v0a1*$^{+/+}$ (WT) and *Atp6v0a1*$^{A512P/A512P}$ (Homo) mice at P10. The regions indicated by dotted boxes in the left two columns of images were expanded in the right two columns. PSD95/VGLUT1 double-positive excitatory synapses (yellow) were decreased in both the hippocampal CA3 and the cerebellar molecular layer in mutant mice. Scale bars, 100 μm. CA3 hippocampal region CA3, MF mossy fiber, IGL internal granule layer, ML molecular layer, EGL external granule layer.
**b** Excitatory synaptic density determined as the ratio of PSD95/VGLUT1 double-positive area to the total area was significantly reduced in both the hippocampus (Hip) and the cerebellum (CB) in *Atp6v0a1*$^{A512P/A512P}$ pups. Data represented as mean values ± standard deviation of six or more measurements. Data in **a** are representative of three independent experiments. Two way ANOVA with Sidak's multiple comparisons test was used (**b**). Source data are provided as a Source data file.

observed in the CA3 regions of *Atp6v0a1*$^{A512P/A512P}$ pups (Fig. 7b). Immunoblot analysis indicated that LC3 of *Atp6v0a1*$^{+/+}$ brain was mainly in its cytosolic form, LC3-I, but in *Atp6v0a1*$^{A512P/A512P}$ pups the membrane-bound form of LC3, LC3-II, was significantly increased in the cortex, which suggests an increase of autophagosomes (Fig. 7c–e). We found no statistically significant difference in LC3-II levels in the hippocampus of *Atp6v0a1*$^{A512P/A512P}$ pups using immunoblotting; however, transmission electron microscopy showed that excessive autophagosomes with double membrane (Fig. 7f, blue), lysosomes with single membrane (Fig. 7f, brown), and rough endoplasmic reticulum were present in mutant CA3 neurons, but not in cerebellum. These results suggest that in *Atp6v0a1*$^{A512P/A512P}$ pups the autophagosome clearance was impaired, and the undegraded autophagosomes and lysosomes were accumulated in the cortical and CA3 pyramidal neurons (Fig. 7g), which led to diffuse distributions of Lamp1 and Lamp2. Thus, the lysosomal dysfunction in *Atp6v0a1*$^{A512P/A512P}$ pups would also have impaired the autophagy function.

**Decreased mTORC1 signal in *Atp6v0a1*$^{A512P/A512P}$ mutants.**
V-ATPase regulates mTORC1 signaling activity[21]. We checked expression of phospho-mTor at serine 2448 (p-mTor) and phospho-S6 (pS6) ribosomal protein as markers for mTORC1 signaling activity. Both in immunohistochemistry and immunoblot, p-mTor level was not changed in brain sections between wild-type and respective *Atp6v0a1*$^{A512P/A512P}$ pups (Fig. 8a–d). The pS6 signal in *Atp6v0a1*$^{+/+}$ pups at P10 was strong in pyramidal neurons in both the upper (II/III) and lower (IV/V) layers of the cortex, and in the neurons in CA1 and CA3 regions of the hippocampus (Fig. 8a), and in cerebellar Purkinje cells (Fig. 8b). However, the signal in *Atp6v0a1*$^{A512P/A512P}$ pups was greatly reduced in the cortex and hippocampus (Fig. 8a), and mildly reduced in Purkinje cells (Fig. 8b). Immunoblotting also showed reduced pS6 signals in these three brain regions at P10 (Fig. 8c, e), and in the hippocampus of P1 pups (Supplementary Fig. 9). Moreover, we examined cytoplasmic/nuclear localization of TFEB, mTORC1 signaling regulator, using Flag-TFEB-mClover2 expression vector in ATP6V0A1-3×HA HEK293FT stable cells. We found no differences in localization of TFEB between wild-type and three mutant expressing cells (Supplementary Fig. 10), which suggests that TFEB activation is not altered by ATP6V0A1 variants. Although the upstream signal of mTORC1 signaling and mTORC1-TFEB pathway was not affected by ATP6V0A1 variants, the reduced level of pS6 in the brain of *Atp6v0a1*$^{A512P/A512P}$ pups suggested impaired mTORC1 activity. Therefore, lysosomal dysfunction may result in cell death, autophagosome/lysosome accumulation, and reduced mTORC1 activity in the brain of *Atp6v0a1*$^{A512P/A512P}$ pups (Fig. 8f).

**Lower neurotransmitter content of synaptic vesicles in *Atp6v0a1*$^{A512P/A512P}$ mice.** The V-ATPase in neurons also plays a crucial role in determining the neurotransmitter content of synaptic vesicles in presynaptic terminals[1]. Protons accumulated by V-ATPases in synaptic vesicles are utilized to incorporate neurotransmitters into the vesicles by various types of vesicular transporters. To examine the effect of the A512P variant on the neurotransmitter content, we first compared miniature excitatory postsynaptic currents (mEPSCs) caused by the release of single glutamate-containing vesicles in primary cortical neurons prepared from *Atp6v0a1*$^{+/+}$ and *Atp6v0a1*$^{A512P/A512P}$ littermate mice. Comparison of a total of 20,000 mEPSC events between *Atp6v0a1*$^{+/+}$ and *Atp6v0a1*$^{A512P/A512P}$ neuron groups clearly indicated a larger proportion of small mEPSCs in *Atp6v0a1*$^{A512P/A512P}$ neurons than in *Atp6v0a1*$^{+/+}$ neurons (Fig. 9a–f). This was not due to the

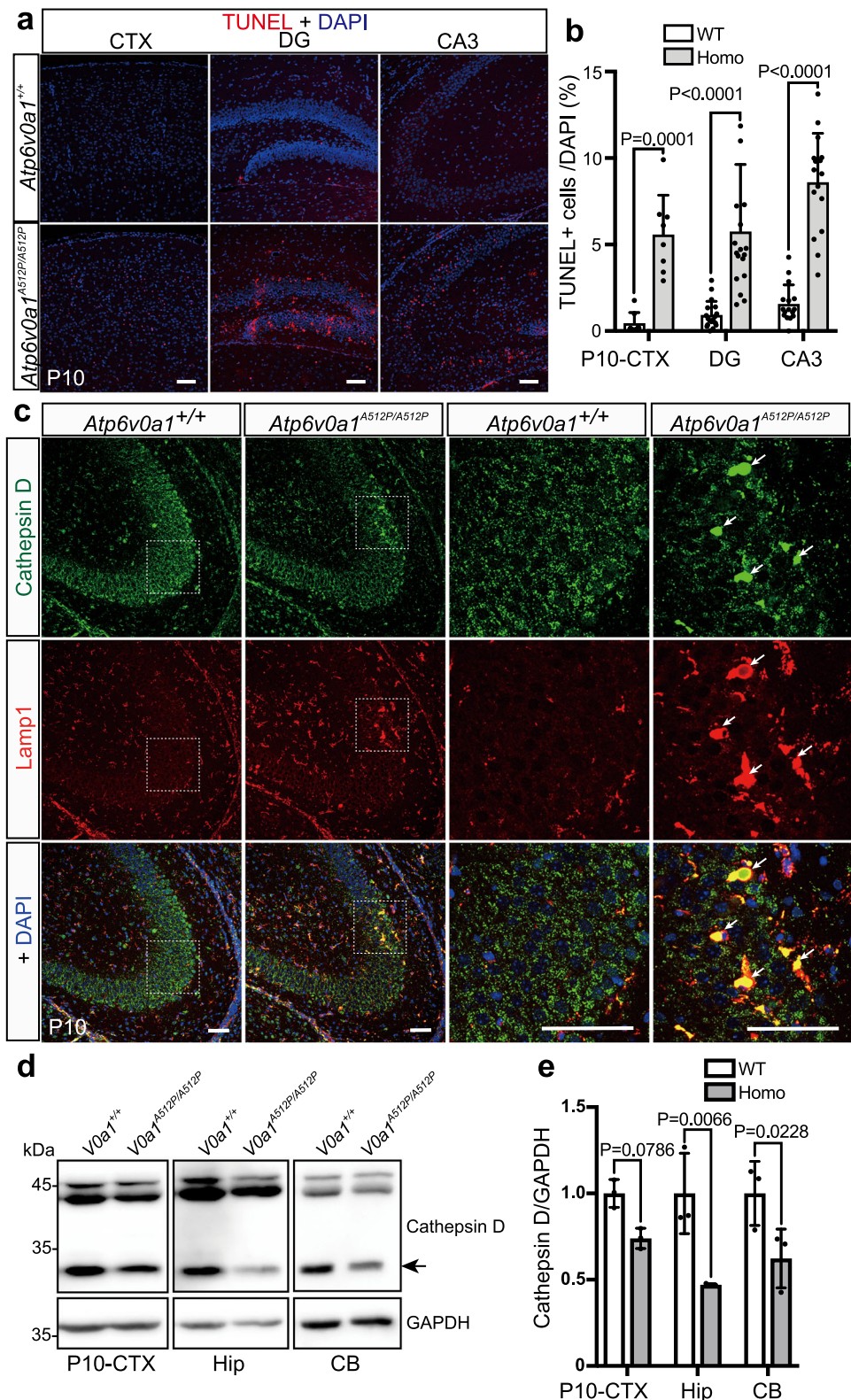

difference in postsynaptic density of the AMPA-type glutamate receptor generating mEPSCs, because the amplitudes of membrane currents evoked by puff application of AMPA were similar between both types of neurons (Supplementary Fig. 11). We also compared miniature inhibitory postsynaptic currents (mIPSCs) caused by the release of single γ-aminobutyric acid (GABA)-containing vesicles in the same way as mEPSCs, and found a significantly larger proportion of small mIPSCs in $Atp6v0a1^{A512P/A512P}$ neurons than in $Atp6v0a1^{+/+}$ neurons (Supplementary Fig. 12a–f). Thus, the neurotransmitter content of synaptic vesicles was indeed lowered in $Atp6v0a1^{A512P/A512P}$ mice, presumably due to the reduced proton pump activity.

**Fig. 6 Increased cell death and impaired lysosomal activity of Atp6v0a1$^{A512P/A512P}$ brain. a** TUNEL-stained coronal sections of the cortex (CTX; left), and the dentate gyrus (DG, middle), and CA3 region (right) of the hippocampus (Hip) in Atp6v0a1$^{+/+}$ (WT) and Atp6v0a1$^{A512P/A512P}$ (Homo) pups at P10. Nuclei were stained with DAPI (blue). **b** Quantification of TUNEL-positive cell numbers in the CTX, DG, and CA3 in Atp6v0a1$^{+/+}$ and Atp6v0a1$^{A512P/A512P}$ pups at P10, showing increased cell death in all these regions in Atp6v0a1$^{A512P/A512P}$ pups. Data represented as mean values ± standard deviation of eight or more counts. **c** Immunostaining of the lysosomal enzyme cathepsin D (green) and the lysosomal membrane marker Lamp1 (red) in the hippocampal CA3 region of Atp6v0a1$^{+/+}$ and Atp6v0a1$^{A512P/A512P}$ pups at P10. Arrows indicate expanded distributions of cathepsin D and Lamp1 proteins. Scale bars, 50 μm. **d** Immunoblot analysis of cathepsin D and GAPDH loading control in the CTX, Hip, and cerebellum (CB) of Atp6v0a1$^{+/+}$ and Atp6v0a1$^{A512P/A512P}$ pups at P10. **e** Quantification of the band intensity of the immunoblot, which shows reduced levels of mature cathepsin D protein (arrow in **d**) in all the three regions of Atp6v0a1$^{A512P/A512P}$ brains. Data represented as mean values ± standard deviation of three independent experiments. Data in **a**, **c**, and **d** are representative of more than three independent experiments. Two way ANOVA with Bonferroni's multiple comparisons test was used (**b**, **e**). Source data are provided as a Source data file.

Moreover, we also found significant rightward shifts of the distributions of interevent intervals of both mEPSC and mIPSC events in the Atp6v0a1$^{A512P/A512P}$ neuron group compared with the Atp6v0a1$^{+/+}$ group (Fig. 9g and Supplementary Fig. 12g). This means that both mEPSCs and mIPSCs occurred less frequently in Atp6v0a1$^{A512P/A512P}$ neurons. Since miniature postsynaptic events are generated by spontaneous release of synaptic vesicles irrespective of neuronal firing activity, the difference would reflect reduced synapse formation onto Atp6v0a1$^{A512P/A512P}$ neurons, as demonstrated for excitatory synapses in the hippocampus and in the cerebellum (Fig. 5).

## Discussion

In this study, we identified de novo and biallelic ATP6V0A1 variants in four individuals with DEE, suggesting that various degrees of impairment of ATP6V0A1 function are associated with neurodevelopmental disorders. The de novo c.2222G>A (p.R741Q) variant is very likely to cause loss-of-function as homozygous mutant mice (Atp6v0a1$^{R741Q/R741Q}$) displayed embryonic lethality as did knockout mice (Atp6v0a1$^{KO/KO}$). Thus, loss-of-function of one of two alleles is associated with DEE. However, the mother of individual 3 who harbors a 50-kb deletion involving exons 1–13 of ATP6V0A1 and the father of individual 4 who harbors a splice site variant (c.196+1G>A) are healthy, suggesting that a simple haploinsufficiency scenario might be less likely. A possible explanation why the heterozygous R741Q variant is associated with DEE might be as follows; the R741Q variant proteins are incorporated into V-ATPase complexes, and they block proton transport without affecting ATP hydrolysis. This results in wasteful consumption of ATP by the mutant V-ATPases, leading to the DEE phenotype. In contrast, the variant with a 50-kb deletion and the splice site variant prevent the production of functional ATP6V0A1 proteins, and they are not incorporated into the complexes. Thus, the remaining intact V-ATPases work properly without wasteful consumption of ATP, and this leads to the normal phenotype.

In this report, we first described on the Atp6v0a1 hypomorphic mutant mice, with a focus on brain phenotypes. V-ATPase family mutant mice have been generated by the International Mouse Phenotyping Consortium (http://www.mousephenotype.org), and knockout mice of many V-ATPase members die during development. Our study also indicated the death of Atp6v0a1$^{KO/KO}$ embryos at an early stage (Supplementary Fig. 4), suggesting that Atp6v0a1 is the core subunit of the V-ATPase family. It has been recently reported that Atp6v0a1 conditional knockout in adult neurons in mice caused a dramatic decrease of hippocampal CA1 neurons[22], while cell death was profound in CA3 neurons in Atp6v0a1$^{A512P/A512P}$ mice. These data suggested stage-specific roles of Atp6v0a1 during brain development.

We demonstrated that both glutamate and GABA contents of synaptic vesicles was lowered in Atp6v0a1$^{A512P/A512P}$ mice (Fig. 9 and Supplementary Fig. 12). To date, pathogenic variants in genes involved in neurotransmission, such as STXBP1, SNAP25, STX1B, DNM1, and PPP3CA are known to cause DEE[23–29], suggesting that disturbance of neurotransmission is associated with DEE. In addition, we demonstrated impaired synaptic connectivity in the hippocampus and cerebellum in Atp6v0a1$^{A512P/A512P}$ pups (Fig. 5). A recent report showed that transient expression of ATP6V1A variants identified in DEE individuals impaired synapse formation in primary hippocampal neurons, which suggests a role for V-ATPase in neuronal development[7]. Our study confirmed the essential role of V-ATPase in synapse formation in vivo and suggested that the impaired synaptic connectivity, including reduced neurotransmitters and synapse formation may contribute to the pathogenesis of DEE caused by ATP6V0A1 variants.

In terms of the functions of ATP6V0A1 in the brain, morpholino knockdown experiments on zebrafish atp6v0a1 revealed a role of atp6v0a1 in the fusion of phagosomes and lysosomes in microglia for digesting apoptotic neurons[30]. The fusion mechanism of phagosomes and lysosomes in microglia, and that of autophagosomes and lysosomes in neurons are similar[30,31]. Moreover, the R755 mutation of drosophila Atp6v0a1 orthologue VHA100-1, which is homologous to Arg741 in human ATP6V0A1, showed the accumulation of autophagosomes in photoreceptor cells[13]. Cortical neurons in Atp6v0a1$^{A512P/A512P}$ mice also indicated the accumulation of an autophagosome marker LC3 and electron microscopy showed excessive autophagosomes and lysosomes in mutant CA3 neurons (Fig. 7). These findings suggest that ATP6V0A1 is evolutionarily essential for the fusion between autophagic vesicles and lysosomes in the degradation machinery (Fig. 8f).

Severe brain atrophy including the cerebellum was observed in two individuals with biallelic ATP6V0A1 variants. Our analysis showed a decrease in brain size of Atp6v0a1$^{A512P/A512P}$ pups at P10. Because mTORC1 signaling promotes cell growth[32], the retardation of cell growth due to the decreased mTORC1 signaling, in addition to the increased cell death due to lysosomal dysfunction, would cause the decrease in brain size in Atp6v0a1$^{A512P/A512P}$ pups. Moreover, Atp6v0a1$^{A512P/A512P}$ pups showed ataxia (Supplementary Fig. 4 and Supplemental Movie). mTORC1 signaling also regulate the dendritic arborization of Purkinje cell[33]. In our analysis, reduced ML and reduced synaptic connectivity of cerebellar parallel fibers were observed in Atp6v0a1$^{A512P/A512P}$ pups (Figs. 4e, f, 5a, and 8b), suggesting that dysfunction of Purkinje cells may be involved in the motor dysfunction. Our histological analysis also revealed the accumulation of LC3 puncta and the expanded distribution of Lamp2 in the cortical pyramidal cells in layer V of Atp6v0a1$^{A512P/A512P}$ pups (Fig. 7a). The axons of these pyramidal neurons project in the corticospinal tract, which is highly connected in the motor execution network[34]. Therefore, the dysfunction of both cerebellar Purkinje cells and cortical pyramidal cells may be involved in the ataxic motor dysfunction of Atp6v0a1$^{A512P/A512P}$ pups.

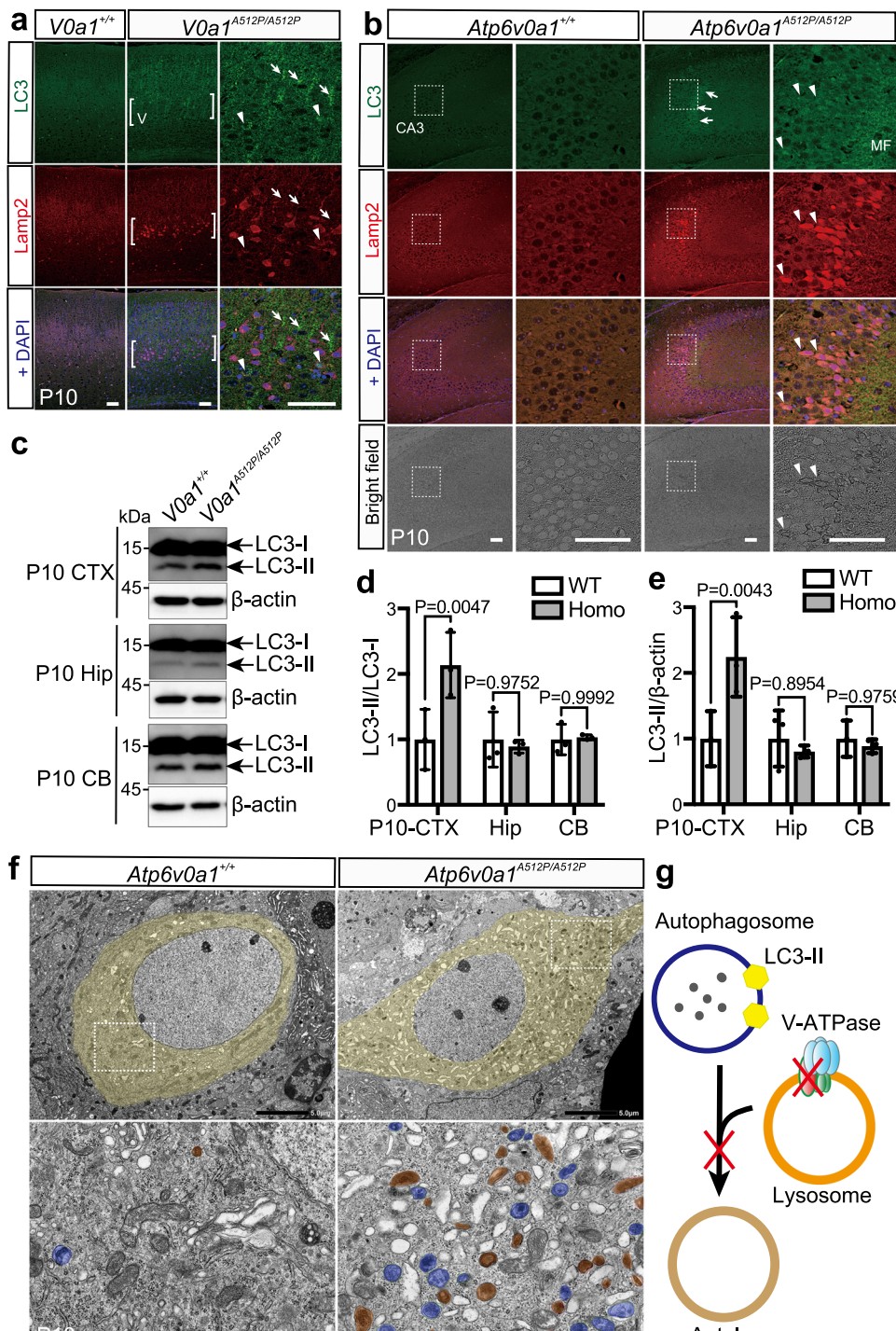

**Fig. 7 Accumulation of autophagosomes and lysosomes in the *Atp6v0a1^{A512P/A512P}* brain. a**, **b** Immunostaining of the autophagosome maker LC3 and the lysosomal membrane marker Lamp2 in coronal sections of the cortex (**a**) and the hippocampal CA3 region (**b**) of *Atp6v0a1^{+/+}* (WT) and *Atp6v0a1^{A512P/A512P}* (Homo) pups at P10. LC3 was accumulated in the cell bodies (arrowheads in **a**) and neurites (arrows in **a**) in layer V of the cortex, and in hippocampal mossy fibers (arrows in **b**) and in the cell bodies of CA3 neurons (arrowheads in **b**), in which expanded Lamp2 distributions were also noticeable. In the bright field images in **b**, healthy CA3 neuronal cells displayed as round-shaped cells with translucent cytoplasm, but abnormal cells with a black boundary were also observed. Scale bars, 50 µm. **c** Immunoblot analysis of LC3 levels in the cortex (CTX), hippocampus (Hip), and cerebellum (CB) of *Atp6v0a1^{+/+}* and *Atp6v0a1^{A512P/A512P}* pups at P10. **d**, **e** Quantification of the band intensity of the immunoblot. The autophagosomal membrane-bound form of LC3, LC3-II, was increased in CTX of *Atp6v0a1^{A512P/A512P}* pups. LC3-II intensity was normalized to the LC3-I (**d**) and β-actin (**e**) intensity, and ratios to the average intensity of WT pups were shown. Data represented as mean values ± standard deviation of three independent experiments. Two way ANOVA with Sidak's multiple comparisons test was used. **f** Electron microscopy images in hippocampal CA3 region of *Atp6v0a1^{+/+}* and *Atp6v0a1^{A512P/A512P}* pups at P10. Neuron of *Atp6v0a1^{A512P/A512P}* showed accumulation of autophagosomes (blue) and lysosomes (brown) in the cytoplasmic region (light yellow). Scale bars, 5 µm (top) and 1 µm (bottom). **g** A schematic representation showing that *Atp6v0a1* variants cause fusion defect of phagosomes and lysosomes and lead to increased level of LC3-II. Data in **a**–**c** are representative of three independent experiments. Electron microscopy data in **f** are representative of two independent experiments. Source data are provided as a Source data file.

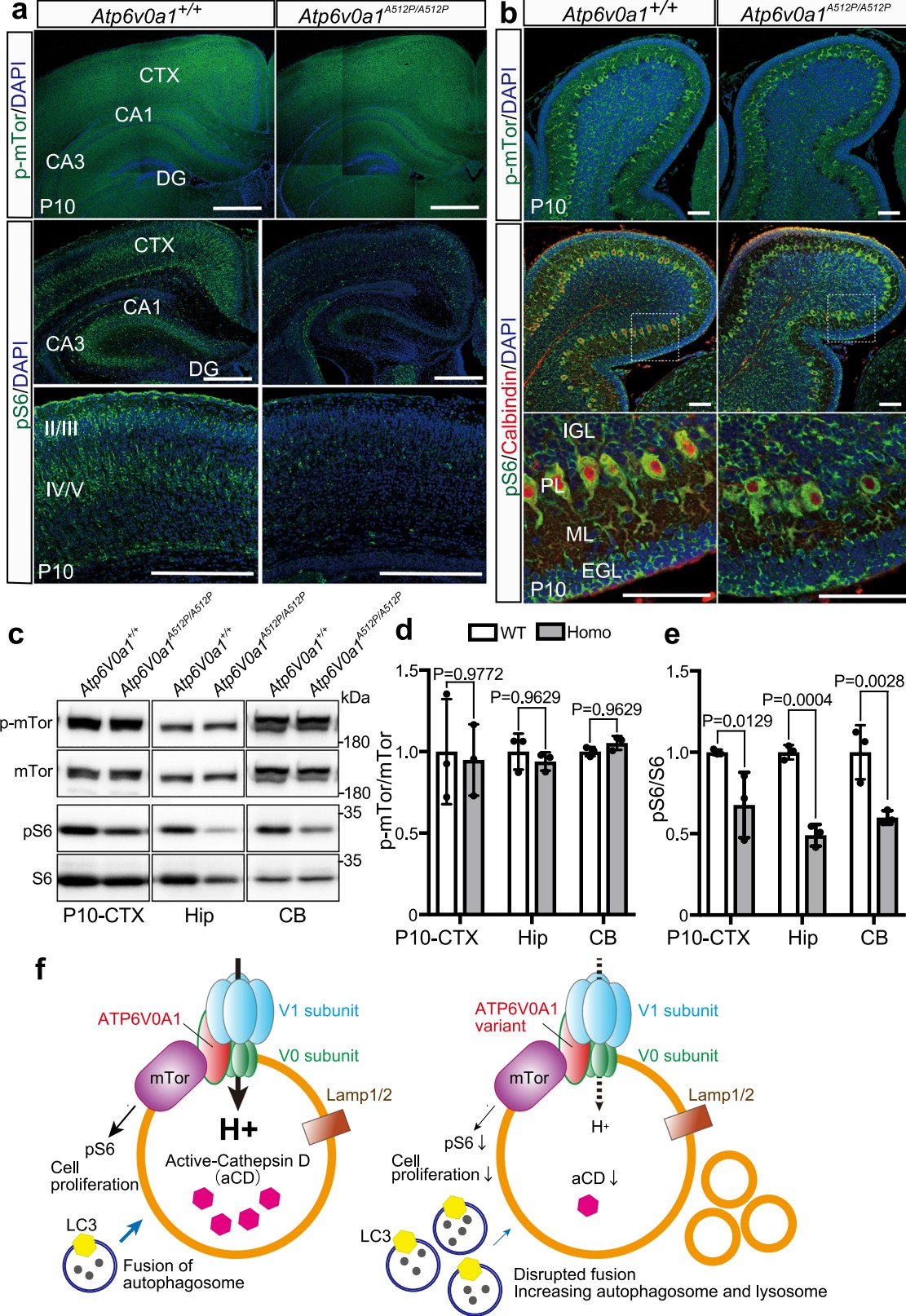

In conclusion, de novo and biallelic *ATP6V0A1* variants were identified in four individuals with DEE. Lysosomal dysfunction resulting in cell death, impaired autophagy, and reduced mTORC1 signaling and synaptic connectivity, and lowered neurotransmitter content of synaptic vesicles were observed in *Atp6v0a1*[A512P/A512P] mice. Our data demonstrated the essential roles of the a1-subunit of the V0 domain of V-ATPase for the brain development in both humans and mice.

## Methods

**Subjects**. A total of 700 individuals with DEEs were analyzed. Clinical information and peripheral blood were obtained from the family members after obtaining their

**Fig. 8 Decreased mTORC1 signaling in *Atp6v0a1*[A512P/A512P] pups. a**, **b** Immunostaining of phospho-mTor (p-mTor) and a downstream target of mTORC1, phospho-S6 (pS6) in the cortex (CTX), and hippocampus (Hip) (**a**) and cerebellum (CB) (**b**) of *Atp6v0a1*[+/+] (WT) and *Atp6v0a1*[A512P/A512P] (Homo) pups at P10. A Purkinje cell marker calbindin was also stained in the cerebellum. DAPI is nuclear marker. The signal of p-mTor was not changed in *Atp6v0a1*[A512P/A512P] pups, but pS6 signal was greatly reduced in the CTX and Hip, and mildly reduced in Purkinje cells. Scale bars, 200 μm in **a** and 50 μm in **b**. **c**, **d** Immunoblot analysis of p-mTor and total mTor, pS6 and total S6 in the CTX, Hip, and CB of *Atp6v0a1*[+/+] and *Atp6v0a1*[A512P/A512P] pups at P10. **d**, **e** The p-mTor and pS6 intensity was normalized to the total mTor and S6 intensity, respectively, and ratios to average intensity of WT pups were shown. Although p-mTor level was not changed (**d**), reduced pS6 signals in three brain regions of P10 pups were recognized (**e**). Data represented as mean values ± standard deviation of three independent experiments. Two way ANOVA with Sidak's multiple comparisons test was used. **f** Intact V-ATPase complex including ATP6V0A1 play roles in lysosomal acidification, lysosome membrane structure, autophagosome clearance, and mTORC1 signal activation (left image). In individuals with *ATP6V0A1* variants and *Atp6v0a1* mutant mice, lysosomal pH is increased, resulting in reduced active cathepsin D, excessive lysosomes (brown circle), and autophagosomes (blue circle) because of fusion defects and reduced mTORC1 signaling (right image). CA1 hippocampal region CA1, CA3 hippocampal region CA3, DG dentate gyrus, L2/3 cortical layers 2 and 3, L4/5 cortical layers 4 and 5, IGL internal granule layer, PL Purkinje cell layer, ML molecular layer, EGL external granule layer. Data in **a–c** are representative of three independent experiments. Source data are provided as a Source data file.

written informed consent, which included a consent to publish. The diagnosis was based on clinical features and characteristic EEG patterns. Experimental protocols were approved by the institutional review boards of Yokohama City University, Hamamatsu University, and Showa University School of Medicine.

**WES**. Genomic DNA from peripheral blood was captured using the Agilent SureSelect Human All Exon V4 or V5 Kit and sequenced on an Illumina HiSeq 2500 with 101 bp paired-end reads. Obtained reads were aligned to the GRCh37 reference genome using NovoAlign (Novocraft Technologies, v3.02). Duplicated reads were removed by Picard (version 1.55), and local realignment and base quality recalibration were performed by Genome Analysis Toolkit (GATK, version 2.7). Variants were identified with the GATK UnifiedGenotyper, and raw variants were filtered out when their parameters met any of the following values: QD < 2.0, MQ < 40.0, FS > 60.0, MQRankSum < −12.5, and ReadPosRankSum < −8.0 for single-nucleotide variants; and QD < 2.0, ReadPosRankSum < −20.0, and FS > 200.0 for insertion/deletions. Final variants were annotated with Annovar[35] for assessing the allele frequency (gnomAD version 2.1.1) and the predictive value of the functional impact of the coding variants (SIFT, Polyphen-2, MutationTaster, CADD v1.3, and M-CAP v1.3). Trio- or quartet-based WES was performed in 211 families. Singleton WES was performed in 489 individuals.

**Single-nucleotide polymorphism array and breakpoint PCR**. One case (individual 3) was studied using the cytogenetics whole-genome 2.7 M array (Affymetrix, Santa Clara, CA). Experimental procedures were performed according to the manufacturer's protocol. Copy number alterations were analyzed using the Chromosome Analysis Suite (Affymetrix) with NA30 (hg19) annotations. The breakpoint junction fragments were confirmed by the following primers: 5′-TCCTGAGGCCTCTTTTCTTG-3′ and 5′-CTGCCACTCACTCATCTCCA-3′.

**Constructs**. A full-length human *ATP6V0A1* was purchased from the Kazusa DNA Research Institute (Chiba, Japan; Flexi ORF Clone ID: FXC26038). Site-directed mutagenesis was performed using a KOD-Plus Mutagenesis kit (Toyobo, Osaka, Japan). All cDNA inserts were verified by Sanger sequencing. ATP6V0A1 was subcloned into the pEF1α-1×HA-hygro vector to express N terminally HA-tagged ATP6V0A1 driven by the EF1α promoter and hygromycin resistance cassette driven by the thymidine kinase promoter. To make neuro2a stable cells expressing ATP6V0A1, ATP6V0A1 cDNA was subcloned into the pEF1α-1×HA-hygro vector to express N-terminal HA-tagged ATP6V0A1 driven by the EF1α promoter, and hygromycin resistance cassette driven by the thymidine kinase promoter. To make HEK293FT stable cells expressing ATP6V0A1, ATP6V0A1 cDNA was subcloned into the pCAG-3×HA-puro vector to simultaneously express C-terminal three HA-tagged ATP6V0A1 and puromycin resistance cassette, both of which were driven by the CAG promoter.

A full-length human TFEB (NM_001167827.3) was purchased from the Kazusa DNA Research Institute (Flexi ORF Clone ID: FXC03423). The cDNA insert was cloned into pGEM-T easy vector (Promega) and verified using Sanger sequencing. TFEB cDNA was subcloned into pEF1α-Flag-mClover2-hygro vector to express N-terminal Flag and C-terminal mClover2-tagged TFEB driven by the EF1α promoter.

**Cell culture**. The neuro2a cell line (ATCC) and HEK293FT (Thermo Fisher Scientific) were cultured with DMEM (Wako, 043-30085), 10% FBS (Biowest, S1820-500), and penicillin–streptomycin (Wako, 161-23201) at 37 °C. Wild-type (WT) and three mutant HA-ATP6V01 expressing neuro2a cells were transfected with polyethylenimine max solution (Polysciences inc. 24765-1, 1 mg/mL) and selected with 600 ng/μL hygromycin B (089-06151, WAKO). HEK293FT cells expressing ATP6V01[WT]-3×HA and mutants were selected with 2 ng/μL puromycin

(160-23151, Wako). For protein stability analysis, stable cell lines were treated with 50 ng/μL cycloheximide (037-20991, Wako) for 0, 2, 4, 6, and 8 h. For glycosylation assays, the stable cell lines were treated with 5 ng/μL tunicamycin (T8153, LKT lab) for 24 h.

**Guide RNA selection using a cell line**. Before we generate mutant mice of *ATP6V0A1*[A512P] and *ATP6V0A1*[R741Q], guide RNA selection for the target site is highly important for the generation of mutant mice. To check the cleavage efficiency for mouse *Atp6v0a1* genomic DNA, all-in-one vector pSpCas9 (BB)-2A-GFP (px458, a gift from Feng Zhang, Addgene #48138) with U6 promoter-driven empty guide RNA, *Streptococcus pyogenes* Cas9, and the GFP reporter gene were used in the N2A cell line. Two and three different guide RNAs were selected for *ATP6V0A1*[A512P] and *ATP6V0A1*[R741Q] knock-in sites, respectively, using the UCSC Genome Browser. After they were transfected to the N2A cell line, collected cells were lysed in 50 mM NaOH/ 0.1 M Tris-HCl pH 7.5 and used for high resolution melting (HRM) analysis using an Eco 48 Real-Time PCR system (PCRmax, Stone, UK). Based on HRM results, guide RNAs which cut target sequences with highest-efficiency were selected for generating mutant mice. A 20-nucleotide target-specific sequence used for generating mutant mice is as follows: CAGTTGAACCCCGCTATCCC for *ATP6V0A1*[A512P] and GGCTGAGGGCCCAGAGTCGC for *ATP6V0A1*[R741Q].

**Generation of the *ATP6V0A1* mutant using an electroporation method**. All animal experiments were performed in accordance with the guidelines of the Physiological Society of Japan and approved by the Institutional Animal Care and Use Committee of the Hamamatsu University School of Medicine. Fertilized one-cell mouse eggs from ICR stain female mice (Japan SLC, Inc., Japan) were collected and incubated in EmbryoMax® KSOM Mouse Embryo Medium (MR-121-D, Merck) until electroporation. CRISPR ribonucleoprotein (RNP) of 0.1 μg/μL Cas9 protein, 0.1 μg/μL crRNA, 0.1 μg/μL tracrRNA (Integrated DNA Technologies), and 0.2 μg/μL single-stranded oligodeoxynucleotides (ssODN, Macrogen Japan) within 130 bp homologies were mixed in Opti-MEM solution (31985062, Thermo Fisher Scientific, Waltham, MA). Ten to 15 eggs were transferred into 1-mm gap electrodes (CUY501P1-1.5, NEPAGENE) with 5 μL CRISPR RNP and ssODN solution, and electroporated with a NEPA21 electroporator (NEPAGENE) under the following condition; poring pulse (40 V, pulse duration 3.5 ms, pulse interval 50 ms, four pulses) and transfer pulse (5 V, 50 ms, 50 ms, five pulses). Electroporated eggs were cultured for 1 day, developed to the two-cell stage and then transferred into the ampulla of a uterine tube in an ICR pseudopregnant mother. Anesthetics Domitor/Midazolam/Betorufare (Nakakita Yakuhin, Japan) and the recovery drug Antisedan (Nakakita Yakuhin, Japan) were used. Guide RNA and ssODN are shown in Supplementary Fig. 3. We confirmed the 227-bp genomic sequence of *Atp6voa1*, which covered the whole of exon 13 and franking 71 and 65-bp intronic sequences. Founder mice were backcrossed for two generations before phenotypes were evaluated.

**Genotyping**. After the body weights of the pups were measured, genomic DNA was extracted from their tail samples. Then HRM analysis was performed to distinguish group 1 (wild-type or homozygote) and group 2 (heterozygote) mice using an Eco 48 Real-Time PCR system (PCRmax). In group 1, PCR products were digested by a suitable restriction enzyme to distinguish wild-type and homozygote pups; SmaI for *Atp6v0a1*[A512P] and *Atp6v0a1*[KO], and PstI for *Atp6v0a1*[R741Q]. The PCR product sizes are as follows: wild-type 283 bp, mutant 151/132 bp for *Atp6v0a1*[A512P]; wild-type 283 bp, mutant 150/132 bp for *Atp6v0a1*[KO]; wild-type 247 bp, and mutant 111/136 bp for *Atp6v0a1*[R741Q].

Genotyping primers used are as follows:
A512P-check-F, TCACTGAGGAATGGAGATAGCTGTAGTCG
A512P-check-R, GCACATGTAAGACCTTCCGTTCAACTC
R741Q-check-F, CTGTGGTTTCTTCCCATGCCAGTTT
R741Q-check-R, ACAGGAGCTGGATGGATTCGAGAAC.

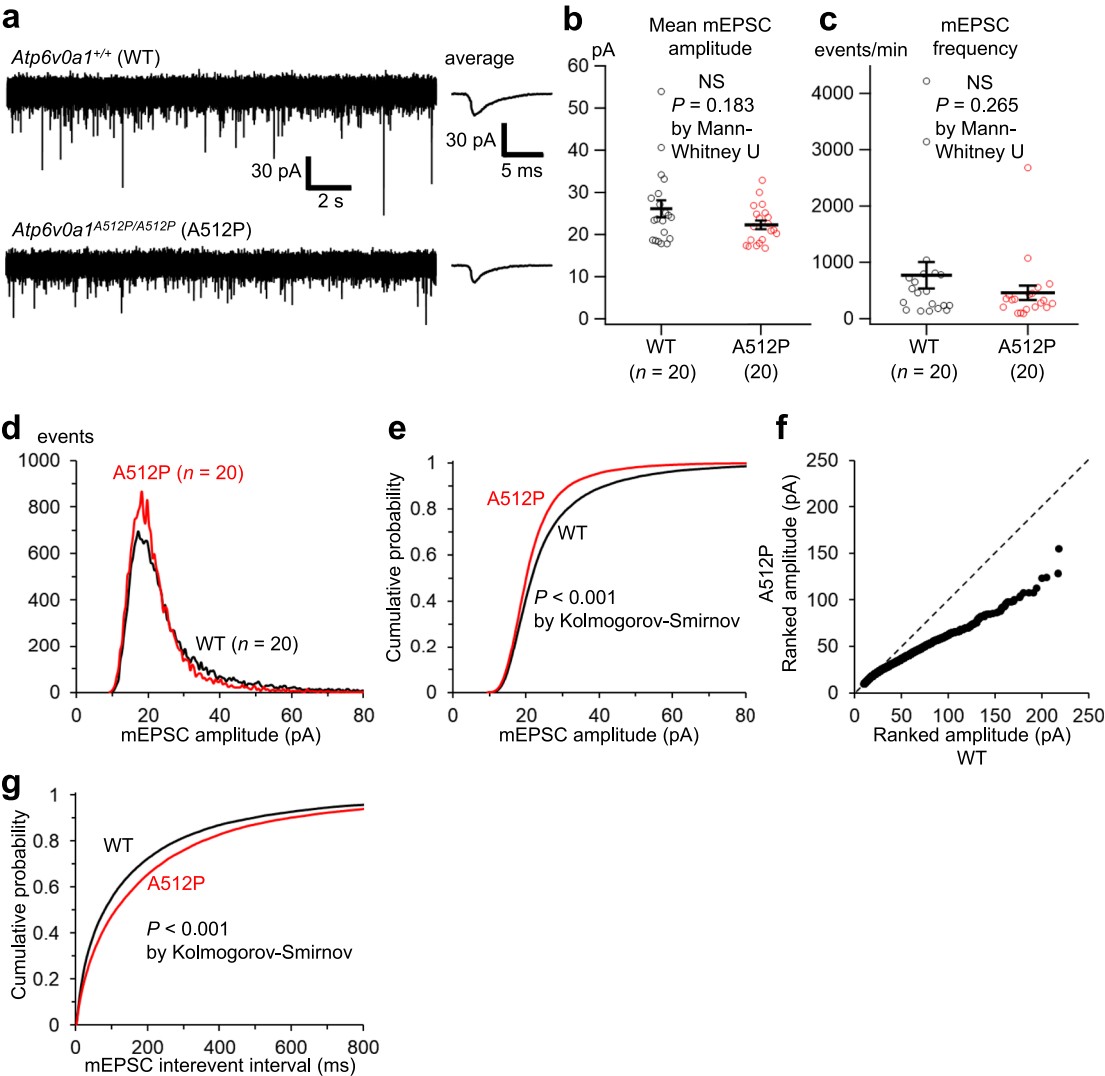

**Fig. 9 Comparison of mEPSCs in *Atp6v0a1*+/+ (WT) and *Atp6v0a1*A512P/A512P (A512P) neurons. a** Representative traces of mEPSCs in WT and A512P neurons. The average of individual mEPSCs in each trace is shown on the right. **b**, **c** The mean amplitude (**b**) and frequency (**c**) of 1000 consecutive mEPSCs in each neuron are plotted and compared. The differences in both parameters between WT and A512P did not reach statistical significance (NS, $P = 0.183$ by Mann–Whitney $U$ test in **b** and $P = 0.265$ by Mann–Whitney $U$ test in **c**). Horizontal line, mean; error bars, s.e.m. **d**, **e** Histograms (**d**) and cumulative distributions (**e**) of the amplitudes of a total of 20,000 mEPSCs in the WT and A512P neuron groups. The bin widths were 0.5 pA in **d** and 0.1 pA in **e**. The plots clearly indicate an increased proportion of small mEPSCs (~20 pA) and a decreased proportion of large mEPSCs (30–50 pA) in the A512P group, compared with the WT group. The distributions were significantly different between WT and A512P ($P < 0.001$ by Kolmogorov–Smirnov test). **f** Ranked mEPSC amplitudes in ascending order in the A512P group were plotted against those in the WT group. The dashed line indicates the level when the amplitude distributions are the same. The slope of the plots was clearly lower than the dashed line as a result of the smaller proportion of large mEPSCs in the A512P group. **g** Cumulative distributions of interevent intervals of mEPSCs, indicating a significantly larger proportion of long intervals in the A512P group ($P < 0.001$ by Kolmogorov–Smirnov test). Source data are provided as a Source data file.

**Lysosomal pH value measurement**. Parental HEK293FT cell line and stable HEK293FT cell lines expressing ATP6V0A1-3×HA were incubated for 1 h at 37 °C with 0.5 mg/mL of a ratiometric pH indicator dye, LysoSensor™ Yellow/Blue dextran (L22460, Thermo Fisher Scientific) in a 96-well plate. Fluorescence was measured with the Synergy H1 Hybrid Multi-Mode Reader (BioTek), using excitation at 355 nm and emission detection at 440 and 535 nm. To obtain a pH calibration curve, cells were resuspended in the MES-buffered calibration solution (115 mM KCl, 5 mM NaCl, 1.2 mM MgSO4 and 25 mM MES (2-morpholi-noethanesulfonic acid, monohydrate, 145224-94-8, DOJINDO) with pH ranging from 3.7 to 7.6) containing ionophores of 10 mM Monensin (CAS:22373-78-0, Cayman) and 10 mM Nigericin (149-07261, Wako). The calibration data set of fluorescence ratios (440 nm/535 nm) were fitted to a linear regression line using the software GraphPad Prism 7. The formula of the line was $Y = 6.383X - 3.635$. For pH measurements, cells were incubated in the MES-buffered solution at pH 7.7 without ionophores. Six independent pH measurements were made for each of the cell lines expressing wild-type and three different mutants of ATP6V0A1 (ATP6V0A1WT-3×HA, ATP6V0A1A512P-3×HA, ATP6V0A1N534D-3×HA, and ATP6V0A1R741Q-3×HA).

**Immunocytochemistry and LysoTracker staining**. N2A and HEK293FT cells were cultured on round coverslips in a 24-well dish. For immunostaining of HEK293FT cells, cells were fixed in 10% formalin/phosphate buffered saline (PBS) solution for 10 min at room temperature (RT) and permeabilized with 0.2% Triton X−100/PBS for 10 min at RT. For co-staining of N2A cells with LysoTracker, cells were first incubated with LysoTracker Red DND-99 (1/10,000, L7528, Thermo Fisher Scientific) for 1 h at 37 °C before fixation, and permeabilization step was omitted. After washing twice with PBS and blocking with 3% bovine serum albumin (BSA)/PBT (PBS + 0.1% Tween-20) for 1 h at RT, cells were incubated overnight at 4 °C with rat anti-HA (1/200, 11867423001, Roche) for both HEK293FT and N2A cells, and also rabbit anti-ATP6V1A (1/500, ab137574, Abcam) and mouse anti-Lamp2 (1/200, ABL-93, DSHB) for HEK293FT cells as primary antibodies. After washing three times with PBS, cells were incubated for 1 h at RT with Alexa Fluor 488 donkey anti-rat IgG (for both HEK293FT and N2A cells), and with Alexa Fluor 546 donkey anti-rabbit IgG and Alexa Fluor 647 goat anti-mouse IgG (for HEK293FT cells, 1/500 dilution, Thermo Fisher Scientific) as secondary antibodies. After washing three times with PBS, cells were stained with the nuclear marker DAPI and mounted with VECTASHIELD Hard Set Mounting

Media (H−1400, Vector). Experiments were repeated twice. Fluorescent images of each cell line were taken from five independent regions using the Leica TSC SP8 confocal microscopy (Leica Microsystems) with identical condition, through a 63× objective lens with or without 4× digital zoom. For LysoTracker stain, the fluorescent intensity of five images of each cell line was measured using Image J software. Ratios to average intensity of untransfected control cells were calculated.

**Righting reflex.** To evaluate righting reflex[36], pups at P3 were placed on their backs and the time taken to right themselves was measured three or four times. The average time was used to evaluate neonatal motor function.

**Immunohistochemistry.** We analyzed three or more independent mice for each genotype (wild-type and $Atp6v0a1^{A512P/A512P}$). The brains of mouse pups at P10 were fixed in 4% formaldehyde (PFA)/PBS solution overnight at 4 °C. After washing with PBS, samples were dehydrated in an ethanol/xylene series and embedded in paraffin. Samples were cut at 5 μm. After deparaffinizing with xylene and washing with serial dilutions of ethanol and PBS, antigen retrieval was performed on sections with 10 mM sodium citrate solution. After blocking with 3% BSA/PBT (PBS + 0.1% Tween-20) for 1 h at RT, sections were incubated overnight at 4 °C with either of the following primary antibodies: mouse anti-calbindin D-28k (1/100, 300, Swant), mouse anti-CNPase (1/100, ab6319, Abcam), mouse anti-GFAP (1/400, G3893, Sigma-Aldrich), rat anti-LAMP1 (1/100, 1D4B, DSHB), mouse anti-LAMP-2 (1/50, ABL-93, DSHB), mouse anti-NeuN (1/100, MAB377, Millipore), rabbit anti-ATP6V0A1 (1/100, gift from Dr. Sato[37]), rabbit anti-ATP6V1A (1/100, ab137574, Abcam), rabbit anti-LC3 (1/100, PM036, MBL), rabbit phosphor-S6 (Ser235/236; 1/100, #4856, Cell Signaling), rabbit phosphor-mTor (Ser2448; 1/100, #5536, Cell Signaling technology), rabbit anti-cathepsin D (1/200, ab75852, Abcam), goat anti-cathepsin D (1/50, sc-6489, Santa Cruz), mouse anti-PSD95 (7E3-1B8; MA1-046, Thermo Fisher Scientific), guinea pig anti-VGLUT1 (AB5905, Millipore), and goat anti-Ibal (1/100, ab5076, Abcam). After washing three times with PBS, cells were incubated for 1 h at RT with Alexa Fluor 488 or 546 donkey anti-mouse IgG, anti-rat IgG, and anti-goat IgG (1/200-1/400 dilution, Thermo Fisher Scientific) as the secondary antibody. After washing three times with PBS, tissues were stained with the nuclear marker DAPI and mounted with Aqueous Mounting media (TA-030-FM, PermaFluor). Mounted slides were imaged using the 10× or 20× objective lens of Leica TSC SP8 confocal microscope (Leica microsystems).

For cell counts of layer V neurons and the measurement of cerebellar layer volumes, coronal sections of the cortices, and sagittal sections of cerebellums from three independent $Atp6v0a1^{+/+}$ and $Atp6v0a1^{A512P/A512P}$ pups were triple stained (calbindin/NeuN/DAPI). Images were taken from more than five cortical regions or lobules II and III of each genotype. Cortical cell count was performed within a 200-μm-wide bin using the analyze particles function of Image J software. The volume of each layer of the cerebellar lobules was calculated using the analyze/measure function of Image J software, and was shown as % ratio of total volume of lobules II and III.

To measure excitatory synapses, coronal sections of CA3 regions, and cerebellar lobules of $Atp6v0a1^{+/+}$ and $Atp6v0a1^{A512P/A512P}$ pups were triple stained (PSD95/VGLT1/DAPI). Images were taken from more than six stratums lucidum and radiatum layers in the hippocampal regions and MLs of the cerebellar lobules for each genotype. PSD95 and VGLUT1 double-positive excitatory synapses within a 50-μm-wide bin were detected using Photoshop software and their area was measured using the analyze particles function of Image J software. Synaptic density was calculated as % of the total bin area and normalized to the average value in wild-type brains.

**Immunoblotting.** We analyzed three independent mice for each genotype (wild-type and $Atp6v0a1^{A512P/A512P}$). Mouse brain tissues were lysed with RIPA buffer (150 mM NaCl, 50 mM Tris-HCl pH 7.4, 1 mM EDTA pH8.0, 1% NP-40, 0.5% sodium deoxycholate, and 0.1% SDS) supplemented with the protease inhibitor cOmplete Mini (11836153001, Roche), and the phosphatase inhibitor PhosSTOP EASYpack (04906837001, Roche). After centrifugation at 9100 × g for 10 min at 4 °C, supernatant samples were mixed with 2× SDS sample buffer, and then analyzed by SDS–PAGE and immunoblotting using a rabbit anti-ATP6V0A1 (1/1000, gift from Dr. Sato), rabbit anti-ATP6V1A (1/1000, ab137574, Abcam), mouse anti-β-actin (1/10,000, MA1-140, Thermo Fisher Scientific), mouse anti-calbindin D-28k (1/1000, 300, Swant), rabbit cathepsin D (1/2000, ab75852, Abcam), mouse anti-Flag (2H8; 1/1000, KO602, Transgenic inc.), mouse anti-GAPDH (1/50,000, 60004-1-Ig, Proteintech), rat anti-HA-Tag (3F10; 1/1000, 11867423001, Roche), rabbit Histone-H3 (1/5000, 17168-AP, Proteintech), rabbit anti-LC3 (1/1000, PM036, MBL), rabbit anti-S6 ribosomal protein (1/1000, #2217, Cell Signaling Technology), rabbit phosphor-S6 (Ser235/236; 1/1000, #4856, Cell Signaling Technology), rabbit mTor (1/2000, #2972, Cell Signaling Technology), and rabbit phosphor-mTor (1/2000, #5536, Cell Signaling Technology). Can get signal immunoreaction enhancer solution (TOYOBO, Osaka, Japan) was used to enhance signal detection. The secondary antibodies were goat anti-rabbit and anti-mouse antibodies conjugated with horseradish peroxidase (1/5000 dilution, Jackson ImmunoResearch, West Grove, PA). Blots were detected using the Clarity Western ECL Substrate (#170-5060, Bio Rad) and the Fusion Chemiluminescence Imaging

System (VILBER). The band intensity was normalized to that of endogenous controls, and the comparison of protein levels between wild-type and $Atp6v0a1^{A512P/A512P}$ was made by normalization to the average protein levels in wild-type brains.

Cytosolic and nuclear fractionation of Flag-TFEB-mClover2 expressing HEK293FT cells was performed using a Cell Fractionation Kit-Standard (ab109719, Abcam), according to the manufacturer's protocol.

**Detection of cell death.** We analyzed three independent mice for each genotype (wild-type and $Atp6v0a1^{A512P/A512P}$). Three serial coronal sections of the brain of each pup were stained using In Situ Cell Death Detection Kit TMR red (12156792910, Sigma-Aldrich). After TUNEL assay, sections were counterstained with the nucleus marker DAPI. The images were taken from four to six regions of each cortex, dentate gyrus, and CA3 region per pup using the 20× objective lens of a SP8 confocal microscope. Counts of TUNEL-positive and DAPI-positive cells using all images were performed using the analyze particles function of Image J software.

**Electron microscopy.** Hippocampus and cerebellum samples of $Atp6v0a1^{+/+}$ and $Atp6v0a1^{A512P/A512P}$ pups at P10 were fixed with 2% glutaraldehyde (Wako) in 0.1 M phosphate buffer (RM102-5L, LSI medicine) for 2 h and postfixed with OsO4 osmium (VIII) oxide (PGM Chemicals LTD.). After samples were dehydrated in an ascending series of ethanol and propylene oxide (165-05026, Wako), samples were embedded in epoxy resin (DMP-30, Quetol 812, DDSA, NMA, NISSHIN EM co., Ltd). Semithin sections were stained with toluidine blue (535-05542, Wako) and Azur II (015-08672, Wako) solutions, and viewed with an Olympus BX53 microscope and DP-72 digital camera. Ultrathin sections were mounted in meshes, stained with lead stain solution (18-0875-2, Sigma-Aldrich), and observed using a JEM-1400plus (JEOL). Digital images of samples were obtained using a CCD camera. Two independent samples of each genotypes were analyzed and abnormal findings were observed in both.

**Electrophysiology.** For electrophysiological analysis of $Atp6v0a1$ mutants, primary cultured cortical neurons were separately prepared from $Atp6v0a1^{+/+}$ and $Atp6v0a1^{A512P/A512P}$ littermate embryos on E16 or E17. Six $Atp6v0a1^{+/+}$ and five $Atp6v0a1^{A512P/A512P}$ littermates on E16 were used for mEPSC recording, and five $Atp6v0a1^{+/+}$ and three $Atp6v0a1^{A512P/A512P}$ littermates on E17 were used for mIPSC recording. Cerebral cortices of the embryos were dissected and treated with TrypLE Express (Thermo Fisher Scientific) enzyme solution at 37 °C for 20 min, and then minced and triturated using a Pasteur pipette. Dissociated neurons were seeded onto poly-L-lysine-coated matrix sheets (Celltight PL Celldesk LF, MS-0113L; Sumitomo Bakelite, Tokyo, Japan) placed at the bottom of individual wells of 24-well plates at a density of $0.5–1 \times 10^{5}$ cells per well. The neurons were incubated at 37 °C in a humidified 5% $CO_2$–95% air environment with Neurobasal Plus culture medium supplemented with B27 Plus, 0.5 mM GlutaMAX-I, and 50 U and 50 μg/mL of penicillin–streptomycin (all from Thermo Fisher Scientific). Osmolality of the medium was 220 mOsm/kg $H_2O$. Experiments were performed after 17–25 days in vitro.

mEPSCs and mIPSCs were recorded in pyramidal-like neurons at 27–30 °C using the whole-cell patch clamp technique with an EPC10 amplifier controlled via Patchmaster software (HEKA Elektronik, Lambrecht, Germany) under a Nikon (Tokyo, Japan) Eclipse Ti-U inverted microscope. The basal extracellular solution contained (in mM): 120 NaCl, 3 KCl, 2 $CaCl_2$, 1 $MgCl_2$, 5 Na-HEPES, 6 HEPES, 5 glucose, and 1 μM tetrodotoxin (pH 7.4, 255 mOsm/kg $H_2O$). For mEPSC recording, 50 μM picrotoxin and 1 μM strychnine were added to the solution for blocking $GABA_A$ and glycine receptors, respectively. For mIPSC recording, 10 μM CNQX and 50 μM AP5 were added for blocking AMPA and NMDA types of glutamate receptors, respectively. Patch pipettes were fabricated from borosilicate glass capillaries using a P-97 puller (Sutter Instrument, Novato, CA). The pipette solution for mEPSC recording contained (in mM): 110 Cs methanesulfonate, 10 CsCl, 2 $MgCl_2$, 10 HEPES, 3 $Na_2ATP$, 0.2 NaGTP, and 0.2 EGTA (pH 7.4, 250 mOsm/kg $H_2O$). The pipette solution for mIPSC recording contained (in mM): 120 CsCl, 2 $MgCl_2$, 11 HEPES, 3 $Na_2ATP$, 0.2 NaGTP, 5 mannitol, and 0.2 EGTA (pH 7.4, 250 mOsm/kg $H_2O$). The liquid junction potentials between the extracellular and pipette solutions were 9.5 mV for mEPSC recording and 2.9 mV for mIPSC recording, and these were corrected online. Pipette resistance was 3–5 MΩ when filled with the solutions. Neurons were voltage-clamped at the holding membrane potential of −70 mV. Series resistance was compensated for by 70%, and records were discarded when the resistance exceeded 20 MΩ. Records were filtered at 5 kHz and sampled at 25 kHz. Fluctuations of baseline currents in records, if any, were removed offline by subtraction of strongly filtered records (with moving average over 200 ms for mEPSCs and over 1 s for mIPSCs) from original records. Individual mEPSC and mIPSC events were analyzed using the threshold search function of the Clampfit software (Molecular Devices, San Jose, CA). The threshold for event detection was set at 2 s.d. of the baseline current noise. One thousand consecutive mEPSC or mIPSC events were analyzed in each neuron. (S)-AMPA (Fujifilm Wako Pure Chemical, Osaka, Japan) dissolved in the extracellular solution at the concentration of 100 μM were focally applied to the soma of a recorded neuron by pressure ejection (100 ms at 1 p.s.i. controlled by a

Narishige (Tokyo, Japan) IM−300 microinjector) via a glass micropipette (tip diameter ~2 µm) positioned within several tens of micrometers of the soma. The position was determined so that the maximum AMPA-induced currents were generated in the neuron.

**Statistics**. Statistical analyses except for electrophysiological data were carried out using GraphPad Prism 7 software. Immunoblotting was performed using three independent samples and assessed by ANOVA (two tailed) followed by Bonferroni's multiple comparisons test. All immunohistochemistry and TUNEL experiments were done in more than three independent replicates and assessed by ANOVA (two tailed) followed by Bonferroni's multiple comparisons test. LysoSenser findings were assessed by ANOVA (two tailed) followed by Dunnett's multiple comparisons test. LysoTracker fluorescent intensity and the body weight of the pups were assessed by ANOVA (one tailed) followed by Dunnett's multiple comparisons test. Graphical data are presented as mean ± s.d. For electrophysiological data, statistical analysis was carried out using IBM SPSS ver.25 software. The normality of data was first assessed with the Kolmogorov–Smirnov test, and the null hypothesis of normality was rejected for all the parameters except the membrane capacitance and the mean mIPSC amplitude in single neurons. Nonparametric comparisons were made with Mann–Whitney's U test, and the distributions of the amplitudes and interevent intervals of mEPSCs and mIPSCs were assessed with the Kolmogorov–Smirnov test. The equality of variances of the data of membrane capacitance and mean mIPSC amplitude was confirmed by Levene's test, and the differences were assessed with Student's t test assuming equal variances. Horizontal and error bars in plots indicate mean ± s.e.m.

**Reporting summary**. Further information on research design is available in the Nature Research Reporting Summary linked to this article.

## Data availability

Exome data are available in the Human Genetic Variation Database [http://www.hgvd.genome.med.kyoto-u.ac.jp/repository/HGV0000012.html]. Raw sequence and individual-level genotype data can be provided via formal collaboration because of the contents of the obtained informed consent. All other data are available within the article and its Supplementary Information Files from the corresponding author upon reasonable request. Data from RefSeq and Uniprot database are following: NM_001130020.1, NM_001167827.3, NP_058616.1, NP_001123492.1, NP_058616.1, NP_990055.1, NP_997837.1, NP_651672.1, NP_014913.3, and Q93050. Source data are provided with this paper.

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

## Acknowledgements

We thank the individuals and their families for participating in this study. We would like to thank Satoshi Sato for the anti-ATP6V0A1 antibody, and Yoko Kumagiri for preparation of electron microscopy samples. We would also like to thank Masumi Tsujimura and Kaori Shibazaki for their technical assistance. This work was supported by grants for: Research on Measures for Intractable Diseases; Comprehensive Research on Disability Health and Welfare, the Strategic Research Program for Brain Science; Initiative on Rare and Undiagnosed Diseases, and IRUD beyond (JP19ek0109297) from the Japan Agency for Medical Research and Development; Grants-in-Aid for Scientific Research on Innovative Areas (Transcription Cycle) and (Non-linear Neuro-oscillology: Towards Integrative Understanding of Human Nature, KAKENHI Grant Number JP15H05872)

from the Ministry of Education, Culture, Sports, Science, and Technology, Japan; Grants-in-Aid for Scientific Research (A) (JP17H01539), (B) (JP16H05160, JP16H05357, JP20H03641), and (C) (JP17K10080, JP17K08534, JP20K07243) from the Japan Society for the Promotion of Science; Creation of Innovation Centers for Advanced Interdisciplinary Research Areas Program in the Project for Developing Innovation Systems from the Japan Science and Technology Agency; grants from the Ministry of Health, Labour, and Welfare; and the Takeda Science Foundation.

## Author contributions

K.A., Naomichi Matsumoto, and H.S. designed and directed the study. K.A., M.K., T.A., Naomichi Matsumoto, and H.S. wrote the manuscript. N.A., J.T., Y.M., K.H., Y.A., R.T., O.E., R.B.-H., E.H., and M.K. collected samples and provided the subjects' clinical information. M.N., C.O., A.T., T.M., S.M., Noriko Miyake, and H.S. performed exome sequencing and Sanger sequencing, H.S. performed single-nucleotide polymorphism array and breakpoint PCR. K.A. performed lysosome acidification assay. K.A. and S.T. generated mutant mice. K.A., H.M., T.M., and H.B. analyzed mutant mice. T.A. and A.F. performed electrophysiological analysis.

## Competing interests

The authors declare no competing interests.
