## [Peer Review File · Nature Communications]

Reviewers' comments:

Reviewer #1 (Remarks to the Author):

This manuscript describes the identification of several mutations in ATP6V0A1 in four patients with developmental and epileptic encephalopathy (DEE). Generation of mice in which ATP6V0A1 was depleted or carried R741Q mutations were embryonic lethal, whereas mutation of A512P resulted in early postnatal mortality. The A512P mutation resulted in reduced ATP6V0A1 levels, deficient lysosomal acidification, autophagosome accumulation, cathepsin leakage from lysosomes, increased neuronal death, reduced mTOR activity and reduced glutamate content of synaptic vesicles.

Overall the manuscript is well-written and the experiments are well-performed and controlled. There are, however, some concerns about novelty. The role of v-ATPase in lysosomal acidification, regulation of mTORC1 activity, and formation of synaptic vesicles is well-established. There is also ample evidence showing the importance of v-ATPase in neuronal health and its implication in different neurodegenerative disorders. Moreover, the authors have recently identified several de novo missense mutations in ATP6V1A in DEE patients, demonstrating a role for v-ATPase in brain development (Fassio et al., *Brain*, 2018; 141: 1703-1718). Therefore, this study does not seem to add much new information on the molecular mechanisms of DEE. Although some possibilities are suggested in the discussion section (e.g. decreased mTOR activity in Purkinje cells, alterations in cortical pyramidal cells), it is still unclear how mutations in v-ATPase affect specific signaling pathways, thus contributing to DEE.

Additional points:

- The serine/threonine kinase mTOR may be part of two different subcomplexes, mTORC1 and mTORC2. Current evidence suggests a role of v-ATPase in the regulation of the mTORC1 complex. Please substitute mTOR by mTORC1 throughout the manuscript.
- Figure 3. Confirm that the different ATP6V0A1 mutants are being incorporated into the v-ATPase complex. Increase the resolution of the images to confirm proper localization to lysosomes.
- Figure 4. Check if the reduction in ATP6V0A1 protein levels results in a concomitant decrease in the expression of other subunits of the v-ATPase complex. It is unclear why the reduction in ATP6V0A1 levels is not appreciated in the images shown in Supplementary Figure 6c.
- Figure 5 and 6. The Lamp2 staining is not convincing. The Lamp2 antibody would be expected to label discrete puncta corresponding to lysosomes, similar to the staining observed for cathepsin D. Instead, the staining is diffused in the cytosol with some accumulation in the nucleus in mutant animals, while it seems practically absent in controls (Figure 5c and 6c).

The increased Lamp2 staining observed in Figures 5 and 6 might be indicative of increased lysosomal biogenesis, likely as result of the cell's attempt to compensate for the defective lysosomal function. Furthermore, the decreased mTORC1 activity observed in ATP6V0A1A512P/A512P mice may lead to increased TFEB activation. It would be interesting to assess whether TFEB activation is altered in

these animals.

- Figure 6c. The LC3II accumulation in the brain of ATP6V0A1A512P/A512P animals is surprisingly modest, mostly considering the well-established role of v-ATPase in autophagosome degradation and the significant reduction in mTORC1 activity shown in Figure 7d. Please include quantification of the LC3II/LC3I and LC3II/actin ratios from several independent experiments to assess significance.

- It is not sufficiently proved that the increase in cathepsin levels is directly responsible for the reported increase in cell death.

- Do the ATP6V0A1A512P/A512P mice show defects in neuronal development and connectivity, as previously suggested for the ATP6V1A mutants (Fassio et al., *Brain*, 2018; 141: 1703-1718)?

Reviewer #2 (Remarks to the Author):

The manuscript by Aoto et al identified mutation of v-ATPase V0a1 subunit in DEE patients and report evidence that V0a1 is important for neuronal cell death and autophagy. Biallelic variants of several v-ATPase subunits genes are linked to human disease; however V0a1 mutations have not previously been reported in humans. Interestingly, Sato et al found two novel V0a1 mutations in DEE case and further demonstrated importance of the V0a1 subunit in neuronal acidification and brain development. Conflicting data were previously published by Lee (2010, 2015) showing that V0a1 is critical for lysosomal acidification whereas Coen et al (2012) claimed that Voa1 is not critical. Aoto et al have clearly demonstrated that V0a1 is essential for neuronal acidification, autophagy, and normal brain development using V0a1 mutant transgenic mice. In this important respect, this manuscript is potentially interesting and the findings would have added significance in helping to resolve a controversy that has influenced how investigators in the AD field have regarded the roles of AD-causing mutations of the gene PSEN1 in mediating pathogenesis in part through the same effects that are demonstrated in this report. Moreover, a similar impairment of V0a1 function in CLN1, a lysosomal storage disorder, is ascribed to failed trafficking of the subunit to lysosomes (Bagh, 2017, *Nat Comm*). In both the PSEN1 and CLN, the clinical phenotype includes a high frequency of seizure activity. The implications of the current work to brain disease are not addressed in the report perhaps because they don't involve a mutation of V0a1; however, they do involve a similar loss of function of the subunit and strong overlaps in the molecular and some clinical phenotypic features. A full consideration of these issues is therefore essential to elevate the interest and importance of the study, given that the phenotypes revealed in the study are rather descriptive and generally unsurprising in light of the expected/known effects of V0a1 loss of function.

Additional major points

The authors found three V0a1 mutation (N741Q, A512P, and N534D) in DEE patients and demonstrate the V0a1 mutation effect on lysosomal acidification using murine neuronal cell line and newly generated transgenic mouse models. Although the hypothesis linking molecular alterations to

clinical symptoms was tested using transgenic mouse, there are additional weaknesses that need to be addressed.

1. Functional studies of the v-ATPase activity is needed. Initially the authors examined the effects of ATP6V0a1 variants using the LysoTracker assay. However, the proper way to examine the effects of ATP6V0a1 variants is measurement of v-ATPase activity. Since commercial kits are available, it should be easy to test whether ATP6V0a1 variants really affect the v-ATPase function. LysoTracker is also not the most rigorous index of lysosomal pH and direct ratiometric measurement to precisely determine pH is recommended. A functional readout of the acidification defect using a fluorescent probe to assay proteolytic activity in lysosomes (or an equivalent outcome measure) should be considered.
2. Author need to explain why levels of ATP6V0a1 variants are decreased in Tg mice (Fig 4G). It should be resolved whether or not protein expression is artificially down-regulated and affects net v-ATPase function or whether mutant ATP6V0a1 is more easily degraded. In vivo measurement may not be feasible but an in vitro cycloheximide assay will answer this question. V0a1 is known to be a glycoprotein. Is there evidence that N741Q, A512P, and N534D mutation also affect glycosylation of the subunit?
3. Cross talk between ATP6V0a1 variants and mTOR pathway is not clear. The authors claim that lysosomal dysfunction mediated by ATP6V0a1 variants reduces mTOR signaling. However, it looks like levels of ATP6V0a1 were reduced in P10 cortex (Fig 4G) whereas levels of phospho-S6 were reduced in P1 cortex (Fig 7C) of the A512P/A512P Tg. Are levels of ATP6V0a1 also reduced in P1 cortex ? To support the authors' claim, the deficits are expected to be present by the same early age.
4. Since the ATP6V0a1 variant was expressed in Ctx/hippo and large dense Cathepsin D accumulation were noticeable in CA3, are there cathepsin D accumulations in Ctx as expected?
5. Given that A512P/A512P Tg mice died within 2 weeks postnatally, is there any abnormality seen in Hetero-Tg mice at older ages?

Minor points

1. Green signal of the IHC is too weak in Fig 5C, so the Lamp2 signal in control mice is difficult to see.
2. Please state how many mice are used for each IHC study and document that the lysosomal abnormality is consistently robust in all Tg mice.

Response to reviewers:

Reviewer #1 (Remarks to the Author):

Thank you for your instructive comments. We revised our manuscript according to your comments. Corrections are shown in blue throughout the manuscript.

- 1. This manuscript describes the identification of several mutations in ATP6V0A1 in four patients with developmental and epileptic encephalopathy (DEE). Generation of mice in which ATP6V0A1 was depleted or carried R741Q mutations were embryonic lethal, whereas mutation of A512P resulted in early postnatal mortality. The A512P mutation resulted in reduced ATP6V0A1 levels, deficient lysosomal acidification, autophagosome accumulation, cathepsin leakage from lysosomes, increased neuronal death, reduced mTOR activity and reduced glutamate content of synaptic vesicles. Overall the manuscript is well-written and the experiments are well-performed and controlled.**

Thank you for your positive comments.

- 2. There are, however, some concerns about novelty. The role of v-ATPase in lysosomal acidification, regulation of mTORC1 activity, and formation of synaptic vesicles is well-established. There is also ample evidence showing the importance of v-ATPase in neuronal health and its implication in different neurodegenerative disorders. Moreover, the authors have recently identified several de novo missense mutations in ATP6V1A in DEE patients, demonstrating a role for v-ATPase in brain development (Fassio et al., Brain, 2018; 141: 1703-1718). Therefore, this study does not seem to add much new information on the molecular mechanisms of DEE. Although some possibilities are suggested in the discussion section (e.g. decreased mTOR activity in Purkinje cells, alterations in cortical pyramidal cells), it is still unclear how mutations in v-ATPase affect specific signaling pathways, thus contributing to DEE.**

From the perspective of genetics, it is important to find a novel responsible gene for human disease and to establish causal relationship between variants and disease phenotype because this would enable us to confirm the solid genetic causes in other patients with unknown etiology. Additionally, we experimentally showed the following important findings in revised experiments:

- 1) The pathogenicity of each variant by demonstrating impairment of lysosomal acidification by pH measurements using LysoSensor Yellow/Blue dextran.
- 2) Impaired activity of *Atp6v0a1* in mice reduced cathepsin D activity and caused accumulation of autophagosomes and lysosomes in neurons of the cortex and hippocampus CA3 region, as shown by immunoblotting, immunohistochemistry and electron microscopy.

3) Reduced synaptic connectivity and lowered neurotransmitter contents (both glutamate and γ -aminobutyric acid) of synaptic vesicles were found in *Atp6v0a1*^{A512P/A512P} mice. Our study first confirmed the essential role of V-ATPase in synapse formation *in vivo* and suggested that the impaired synaptic connectivity, including reduced neurotransmitters and synapse formation, may contribute to the pathogenesis of DEE.

Although a full view of pathological mechanism should be further clarified, we believe our findings, that is, the identification a novel responsive gene, pathogenicity of variants and abnormalities related to V-ATPase dysfunction, are worth publishing in *Nature Communications*.

3. - The serine/threonine kinase mTOR may be part of two different subcomplexes, mTORC1 and mTORC2. Current evidence suggests a role of v-ATPase in the regulation of the mTORC1 complex. Please substitute mTOR by mTORC1 throughout the manuscript.

Thank you for your important suggestion. We substituted mTOR with mTORC1 throughout the manuscript.

4. - Figure 3. Confirm that the different ATP6V0A1 mutants are being incorporated into the v-ATPase complex. Increase the resolution of the images to confirm proper localization to lysosomes.

We clarified the localization of ATP6V0A1 and ATP6V1A (A-subunit of the V1 domain) protein to lysosome in high magnification view using HEK293FT cell lines stably expressing ATP6V0A1-3xHA. ATP6V0A1^{WT}-3xHA and ATP6V1A were partially co-localized with Lamp2, which suggests that assembled V-ATPase complex including ATP6V0A1 and ATP6V1A was localized in the lysosome. The localization of three ATP6V0A1 mutants was similar to that of wild-type ATP6V0A1, which suggests that lysosomal localization was not changed among wild-type and three mutant ATP6V0A1. We show these data in revised Fig. 3a and revised the text as follows.

“To examine the effects of *ATP6V0A1* variants on the cellular localization and lysosomal acidification, we established stable human embryonic kidney cell lines (HEK293FT) expressing C-terminal 3xHA-tagged human ATP6V0A1 (ATP6V0A1-3xHA) and stable neuroblastoma-2A (N2A) cell lines expressing N-terminal HA-tagged human ATP6V0A1 (HA-ATP6V0A1). Co-staining with antibodies for HA-tag, ATP6V1A (A-subunit of the V1 domain) and lysosomal marker Lamp2 revealed that wild-type ATP6V0A1 (ATP6V0A1^{WT}-3xHA) was partially co-localized with ATP6V1A and Lamp2, which suggests that assembled V-ATPase complex including ATP6V0A1 and ATPV1A was localized in the lysosome (Fig. 3a). The localizations of three ATP6V0A1 mutants (ATP6V0A1^{A512P}-3xHA, ATP6V0A1^{N534D}-3xHA and ATP6V0A1^{R741Q}-3xHA) were similar to those of wild-type ATP6V0A1 (Fig. 3a).” (page 7, paragraph 2)

5. - Figure 4. Check if the reduction in ATP6V0A1 protein levels results in a concomitant decrease in the expression of other subunits of the v-ATPase complex. It is unclear why the reduction in ATP6V0A1 levels is not appreciated in the images shown in Supplementary Figure 6c.

According to your comments, we checked the protein level of ATP6V1A (A-subunit of the V1 domain) in *Atp6v0a1*^{A512P/A512P} mice at postnatal day 10 using immunoblotting. The Atp6v1a protein level was significantly decreased in cerebellum of *Atp6v0a1*^{A512P/A512P} mutant mice, but not in the cortex and hippocampus, which suggests that the A512P variant could variably affect the levels of other subunits of the V-ATPase complex. Additionally, we examined the protein stability of ATP6V0A1 *in vitro* using ATP6V0A1-3xHA stable cell lines according to reviewer#2's comments. However, we could not find any obvious differences among protein levels of wild-type and three mutant ATP6V0A1-3xHA during 8 hours' treatment with the protein synthesis inhibitor cycloheximide. These data suggest that the A512P variant could affect protein levels *in vivo*, but our *in vitro* test is not sensitive enough to detect this difference during 8-hour periods. We show these data in revised Fig. 4g-I and Supplementary Fig. 7 and revised the text as follows.

“The Atp6v0a1 protein levels were significantly decreased in the cortex, hippocampus and cerebellum of *Atp6v0a1*^{A512P/A512P} pups at P10 (Fig. 4g, h), and a tendency towards reduced level was observed as early as at P1 (Supplementary Fig. 6d). We also examined protein level of another major V-ATPase subunit, Atp6v1a (A-subunit of the V1 domain), and found that Atp6v1a level was significantly decreased in cerebellum of *Atp6v0a1*^{A512P/A512P} pups (Fig. 4g, i). These data suggest that *Atp6v0a1*^{A512P} may affect protein stability of Atp6v0a1, which could affect the protein levels of other V-ATPase subunits *in vivo*. To test this possibility, we performed *in vitro* cycloheximide assay using ATP6V0A1-3xHA HEK293FT stable cell lines (Supplementary Fig. 7a, b). ATP6V0A1^{WT}-3xHA proteins were stable and only 20% reduction was observed after 8 h cycloheximide treatment. In this condition, we found no differences among protein levels of wild-type and three mutant ATP6V0A1-3xHA. These data suggest that the A512P variant could affect protein levels *in vivo*, but our *in vitro* test may be not sensitive enough to detect this difference during 8 h periods. ATP6V0A1 proteins undergo glycosylation¹⁵, and glycosylation difference might contribute to the pathogenicity of three variants. However, the glycosylation of three mutants was not changed compared with that of wild-type HA-ATP6V0A1 (Supplementary Fig. 7c).” (page 10, paragraph 1)”

Concerning ATP6V0A1 levels in Supplementary Figure 6c, decreased ATP6V0A1 levels are obvious in the cortex and hippocampus (upper two panels). In the cerebellum, ATP6V0A1 levels are clearly decreased in Purkinje cells (arrowheads). Please note that marginal staining of cerebellum in *Atp6v0a1*^{A512P/A512P} brain is considered nonspecific staining.

- 6. Figure 5 and 6. The Lamp2 staining is not convincing. The Lamp2 antibody would be expected to label discrete puncta corresponding to lysosomes, similar to the staining observed for cathepsin D. Instead, the staining is diffused in the cytosol with some accumulation in the nucleus in mutant animals, while it seems practically absent in controls (Figure 5c and 6c).**
- and**
- 7. The increased Lamp2 staining observed in Figures 5 and 6 might be indicative of increased lysosomal biogenesis, likely as result of the cell's attempt to compensate for the defective lysosomal function.**

We greatly appreciate your important suggestions. Using another set of antibodies for cathepsin D (ab75852, Abcam) and lysosomal membrane (Lamp1: 1D4B, DSHB), we detected discrete puncta of lysosomes in the wild-type pups (Fig. 6c). Using these antibodies, we confirmed that lysosome membranes were diffusely stained in the cytosol in a subset of hippocampal CA3 neurons in *Atp6v0a1*^{A512P/A512P} brain. Additionally, we examined the lysosomes in hippocampal CA3 neurons of *Atp6v0a1*^{A512P/A512P} pups using electron microscopy and found that both lysosomes and autophagosomes were increased in mutant neurons (Fig. 7f). Because immunoblotting showed that the mature form of cathepsin D was significantly reduced (Fig. 6d), we concluded that impaired lysosomal activity may increase numbers of undegraded autophagosomes and lysosomes, resulting in diffuse staining by Lamp1 and Lamp2 antibodies in mutants (Fig. 6 and 7). We show these data in revised Figs. 6 and 7 and Supplementary Fig. 8 and revised the text as follows.

“However, in *Atp6v0a1*^{A512P/A512P} pups, large dense cathepsin D accumulations were noticeable and they were co-localized with abnormally dispersed distributions of lysosome-associated membrane protein 1 (Lamp1) in the hippocampal CA3 region and dentate gyrus, and they were relatively few in cortex at P10 (Fig. 6c and Supplementary Fig. 8). Immunoblotting revealed that the mature form of cathepsin D was significantly reduced in hippocampus and cerebellum (Fig. 6d, arrow, e). These results suggested that lysosomal activity was impaired in *Atp6v0a1*^{A512P/A512P} pups, which caused abnormal cellular distribution of lysosomes and may be related to neuronal death.” (page 11, paragraph 1)

“We found no statistically significant difference in LC3-II levels in the hippocampus of *Atp6v0a1*^{A512P/A512P} pups using immunoblotting; however, transmission electron microscopy showed that excessive autophagosomes with double membrane (Fig. 7f, blue), lysosomes with single membrane (Fig. 7f, brown) and rough endoplasmic reticulum were present in mutant CA3 neurons, but not in cerebellum. These results suggest that in *Atp6v0a1*^{A512P/A512P} pups the autophagosome clearance was impaired and the undegraded autophagosomes and lysosomes were accumulated in the

cortical and CA3 pyramidal neurons (Fig. 7g), which led to diffuse distributions of Lamp1 and Lamp2.” (pages 11 to 12)

- 8. Furthermore, the decreased mTORC1 activity observed in ATP6V0A1^{A512P/A512P} mice may lead to increased TFEB activation. It would be interesting to assess whether TFEB activation is altered in these animals.**

Thank you for your suggestion. Endogenous *Tfeb* is supposed to be weakly expressed in mouse brain based on the Allen Brain Atlas (<https://portal.brain-map.org>). Additionally, TFEB antibody using immunohistochemistry for mouse brain has not been reported. Therefore, we performed transient transfection experiments using HEK293FT stable cell lines expressing C-terminally 3xHA-tagged human wild-type and mutant ATP6V0A1 (ATP6V0A1-3xHA) with a human TFEB (Flag-TFEB-mClover2) expression vector in which N terminal flag tagged and C terminal mClover2 (a variant of GFP) tagged TFEB is produced. We measured nuclear/cytoplasmic localization ratio, but it was not different between wild-type and three mutants (Supplementary Fig. 10), which suggests that TFEB activation is not altered by ATP6V0A1 variants. We show this result in revised Supplementary Fig. 10 and revised the text as follows.

“Moreover, we examined cytoplasmic/nuclear localization of TFEB, mTORC1 signaling regulator, using Flag-TFEB-mClover2 expression vector in ATP6V0A1-3xHA HEK293FT stable cells. We found no differences in localization of TFEB between wild-type and three mutant expressing cells (Supplementary Fig. 10), which suggests that TFEB activation is not altered by ATP6V0A1 variants. Although the upstream signal of mTORC1 signaling and mTORC1-TFEB pathway was not affected by ATP6V0A1 variants, the reduced level of pS6 in the brain of *Atp6v0a1*^{A512P/A512P} pups suggested impaired mTORC1 activity.” (page 12, paragraph 2)

- 9. - Figure 6c. The LC3II accumulation in the brain of ATP6V0A1A512P/A512P animals is surprisingly modest, mostly considering the well-established role of v-ATPase in autophagosome degradation and the significant reduction in mTORC1 activity shown in Figure 7d. Please include quantification of the LC3II/LC3I and LC3II/actin ratios from several independent experiments to assess significance.**

We performed immunoblot analysis and measured LC3-II/LC3-I and LC3-II/ β -actin ratios using the cortex, hippocampus and cerebellum of three wild-type and three mutant pups at P10. The LC3-II/LC3-I and LC3-II/ β -actin ratios were significantly increased in cortex, but not in hippocampus and cerebellum. We show these data in revised Fig. 7c-e and revised the text as follows.

“Immunoblot analysis indicated that LC3 of *Atp6v0a1*^{+/+} brain was mainly in its cytosolic form, LC3-I, but in *Atp6v0a1*^{A512P/A512P} pups the membrane-bound form of LC3, LC3-II, was significantly increased in the cortex, which suggests an increase of autophagosomes (Fig. 7c-e).” (page 11, paragraph 2)

10. - It is not sufficiently proved that the increase in cathepsin levels is directly responsible for the reported increase in cell death.

In Fig. 5c (revised Fig. 6c), we misunderstood the relationship between cathepsin D and cell death pathway in *ATP6V0A1*^{A512P/A512P} mutant brain. During revision, we obtained evidence of decreased cathepsin D activity by immunoblotting using new cathepsin D antibodies (ab75852, Abcam) and accumulation of lysosomes by electron microscopy. These results indicated that the lysosomal enzymatic activity in mutant brain was decreased and that lysosomal dysfunction may induce neuronal cell death. These data are shown in revised Fig. 6d, e and we revised the text as follows.

“Immunoblotting revealed that the mature form of cathepsin D was significantly reduced in hippocampus and cerebellum (Fig. 6d, arrow, e). These results suggested that lysosomal activity was impaired in *Atp6v0a1*^{A512P/A512P} pups, which caused abnormal cellular distribution of lysosomes and may be related to neuronal death.” (page 11, paragraph 1)

11. - Do the ATP6V0A1^{A512P/A512P} mice show defects in neuronal development and connectivity, as previously suggested for the ATP6V1A mutants (Fassio et al., Brain, 2018; 141: 1703-1718)?

We greatly appreciated your valuable suggestion. In a published paper regarding *de novo* mutations in *ATP6V1A* (Fassio et al., Brain, 2018), pathological variants of *ATP6V1A* dramatically impaired the formation and maintenance of excitatory synapses that were demonstrated by double staining with the presynaptic marker VGLUT1 and the postsynaptic marker Homer1 in hippocampal primary neuron culture. Therefore, we examined the neuronal connection in hippocampal CA3 regions and cerebellar lobules of wild-type and *ATP6V0A1*^{A512P/A512P} mice by double staining against the presynaptic marker VGLUT1 and the postsynaptic marker PSD95. VGLUT1 and PSD95 double-positive excitatory synapses were dramatically reduced in both the hippocampal and cerebellar regions in *Atp6v0a1*^{A512P/A512P} pups at P10. Reduced synapse formation was also suggested by miniature excitatory and inhibitory postsynaptic currents (Fig. 9g and Supplementary Fig. 12g). Therefore, *ATP6V0A1*^{A512P/A512P} mice showed defects in neuronal development and connectivity, first demonstrating the essential role of V-ATPase in synapse formation *in vivo*. We show these data in revised Fig. 5, Fig. 9g and Supplementary Fig. 12g and revised the text as follows.

“The cerebellums of *Atp6v0a1*^{A512P/A512P} pups had normal lobules I-X, but showed a reduced molecular layer (ML) (Fig. 4e, f), suggesting that dendrite development of Purkinje cells would be impaired. Moreover, to examine the neuronal connection in wild-type and mutant mice, double immunostaining of the presynaptic marker VGLUT1 and the postsynaptic marker PSD95 was performed in hippocampal CA3 regions and cerebellar lobules. Excitatory synapses are provided by mossy fibers in stratum lucidum and stratum radiatum in the hippocampal regions and by parallel fibers in molecular layers of the cerebellar lobules. VGLUT1 and PSD95 double-positive excitatory synapses were dramatically reduced in both the hippocampal and cerebellar regions in *Atp6v0a1*^{A512P/A512P} pups at P10 (Fig. 5). Thus, *ATP6V0A1*^{A512P/A512P} mice showed defects in neuronal development and synapse formation.” (page 9, paragraph 1)

“Moreover, we also found significant rightward shifts of the distributions of interevent intervals of both mEPSC and mIPSC events in the *Atp6v0a1*^{A512P/A512P} neuron group compared with the *Atp6v0a1*^{+/+} group (Fig. 9g and Supplementary Fig. 12g). This means that both mEPSCs and mIPSCs occurred less frequently in *Atp6v0a1*^{A512P/A512P} neurons. Since miniature postsynaptic events are generated by spontaneous release of synaptic vesicles irrespective of neuronal firing activity, the difference would reflect reduced synapse formation onto *Atp6v0a1*^{A512P/A512P} neurons, as demonstrated for excitatory synapses in the hippocampus and in the cerebellum (Fig. 5).” (page 13, paragraph 2)

“We demonstrated that both glutamate and GABA contents of synaptic vesicles was lowered in *Atp6v0a1*^{A512P/A512P} mice (Fig. 9 and Supplementary Fig. 12). To date, pathogenic variants in genes involved in neurotransmission such as *STXBP1*, *SNAP25*, *STX1B*, *DNM1* and *PPP3CA* are known to cause DEE^{23, 24, 25, 26, 27, 28, 29}, suggesting that disturbance of neurotransmission is associated with DEE. Additionally, we demonstrated impaired synaptic connectivity in the hippocampus and cerebellum in *Atp6v0a1*^{A512P/A512P} pups (Fig. 5). A recent report showed that transient expression of ATP6V1A variants identified in DEE patients impaired synapse formation in primary hippocampal neurons, which suggests a novel role for V-ATPase in neuronal development⁷. Our study confirmed the essential role of V-ATPase in synapse formation *in vivo* and suggested that the impaired synaptic connectivity including reduced neurotransmitters and synapse formation may contribute to the pathogenesis of DEE caused by *ATP6V0A1* variants.” (page 15, paragraph 2)

Reviewer #2 (Remarks to the Author):

Thank you for your instructive comments. We revised our manuscript according to your comments. Corrections are shown in blue throughout the manuscript.

- 1. The manuscript by Aoto et al identified mutation of v-ATPase V0a1 subunit in DEE patients and report evidence that V0a1 is important for neuronal cell death and autophagy. Biallelic variants of several v-ATPase subunits genes are linked to human disease; however V0a1 mutations have not previously been reported in humans. Interestingly, Sato et al found two novel V0a1 mutations in DEE case and further demonstrated importance of the V0a1 subunit in neuronal acidification and brain development. Conflicting data were previously published by Lee (2010, 2015) showing that V0a1 is critical for lysosomal acidification whereas Coen et al (2012) claimed that Voa1 is not critical. Aoto et al have clearly demonstrated that V0a1 is essential for neuronal acidification, autophagy, and normal brain development using V0a1 mutant transgenic mice. In this important respect, this manuscript is potentially interesting and the findings would have added significance in helping to resolve a controversy that has influenced how investigators in the AD field have regarded the roles of AD-causing mutations of the gene PSEN1 in mediating pathogenesis in part through the same effects that are demonstrated in this report. Moreover, a similar impairment of V0a1 function in CLN1, a lysosomal storage disorder, is ascribed to failed trafficking of the subunit to lysosomes (Bagh, 2017, Nat Comm). In both the PSEN1 and CLN, the clinical phenotype includes a high frequency of seizure activity. The implications of the current work to brain disease are not addressed in the report perhaps because they don't involve a mutation of V0a1; however, they do involve a similar loss of function of the subunit and strong overlaps in the molecular and some clinical phenotypic features. A full consideration of these issues is therefore essential to elevate the interest and importance of the study, given that the phenotypes revealed in the study are rather descriptive and generally unsurprising in light of the expected/known effects of V0a1 loss of function.**

Thank you for your positive assessments and for your important suggestions about PSEN1-ATP6V0A1 association. Lee et al. demonstrated that PSEN1 regulates ATP6V0A1 function and localization to lysosomes via its glycosylation (Lee et al., Cell 2010). According their report, we examined glycosylation and localization of wild-type ATP6V0A1 and three mutants.

To examine the glycosylation of ATP6V0A1, we treated HA-ATP6V0A1 expressing N2A cells with protein glycosylation inhibitor, tunicamycin (TUN, 5 μ g/ml, T8153, CAY) for 24 h and performed immunoblot analysis with HA and GAPDH antibodies. All HA-ATP6V0A1 wild-type and mutants showed two bands of glycosylated mature (116kDa) and immature (100kDa) protein in untreated condition, but bands of glycosylated mature protein disappeared during TUN treatment. Thus, three mutant proteins were not affected in glycosylation. We also clarified the localization of

ATP6V0A1 and ATP6V1A (A-subunit of the V1 domain) protein to lysosome in high magnification view using HEK293FT cell lines stably expressing ATP6V0A1-3xHA. ATP6V0A1^{WT}-3xHA and ATP6V1A were partially co-localized with Lamp2, which suggests that assembled V-ATPase complex including ATP6V0A1 and ATP6V1A was localized in the lysosome. The localization of three ATP6V0A1 mutants was similar to that of wild-type ATP6V0A1, which suggests that lysosomal localization was not changed among wild-type and three mutants. Therefore, it is likely that three mutations did not specifically affect PREN1-ATP6V0A1 association. We show the new experimental data in revised Fig. 3a and Supplementary Fig. 7c and revised the text as follows.

“To examine the effects of *ATP6V0A1* variants on the cellular localization and lysosomal acidification, we established stable human embryonic kidney cell lines (HEK293FT) expressing C-terminal 3xHA-tagged human ATP6V0A1 (ATP6V0A1-3xHA) and stable neuroblastoma-2A (N2A) cell lines expressing N-terminal HA-tagged human ATP6V0A1 (HA-ATP6V0A1). Co-staining with antibodies for HA-tag, ATP6V1A (A-subunit of the V1 domain) and lysosomal marker Lamp2 revealed that wild-type ATP6V0A1 (ATP6V0A1^{WT}-3xHA) was partially co-localized with ATP6V1A and Lamp2, which suggests that assembled V-ATPase complex including ATP6V0A1 and ATPV1A was localized in the lysosome (Fig. 3a). The localizations of three ATP6V0A1 mutants (ATP6V0A1^{A512P}-3xHA, ATP6V0A1^{N534D}-3xHA and ATP6V0A1^{R741Q}-3xHA) were similar to those of wild-type ATP6V0A1 (Fig. 3a).” (page 7, paragraph 2)

“ATP6V0A1 proteins undergo glycosylation¹⁵, and glycosylation difference might contribute to the pathogenicity of three variants. However, the glycosylation of three mutants was not changed compared with that of wild-type HA-ATP6V0A1 (Supplementary Fig. 7c).” (page 10, paragraph 1)

Concerning the novelty of our paper, it is important to find a novel responsible gene for human disease and to establish a causal relationship between variants and human disease/phenotype because it enables us to confirm genetic causes in patients with unknown etiology. Additionally, we experimentally showed several important findings in the revised experiments as described below.

2. The authors found three V0a1 mutation (N741Q, A512P, and N534D) in DEE patients and demonstrate the V0a1 mutation effect on lysosomal acidification using murine neuronal cell line and newly generated transgenic mouse models. Although the hypothesis linking molecular alterations to clinical symptoms was tested using transgenic mouse, there are additional weaknesses that need to be addressed.

In this revision, we examined synaptic connections in *Atp6v0a1*^{A512P/A512P} mice to address the mechanisms of DEE caused by *ATP6V0A1* variants. The excitatory synapses in hippocampal CA3 regions and cerebellar lobules of wild-type and mutant mice were detected by double staining against

the presynaptic marker VGLUT1 and the postsynaptic marker PSD95. Notably, VGLUT1 and PSD95 double-positive excitatory synapses were dramatically reduced in both the hippocampal and cerebellar regions in *Atp6v0a1^{A512P/A512P}* pups at P10 (Fig. 5). These results suggest that excitatory synapses are impaired in terms of synapse formation (Fig. 5) and the glutamate content of synaptic vesicles (Fig. 9).

To examine whether inhibitory synapses are affected, we measured miniature inhibitory postsynaptic currents caused by the release of single γ -aminobutyric acid (GABA)-containing vesicles. The GABA content of synaptic vesicles were reduced (Supplementary Fig. 12), which suggests that inhibitory synapses were also impaired. In addition, reduced excitatory and inhibitory synapse formation was suggested by miniature excitatory and inhibitory postsynaptic currents (Fig. 9g and Supplementary Fig. 12g). Therefore, our study first confirmed the essential role of V-ATPase in synapse formation *in vivo* and suggested that the impaired synaptic connectivity including reduced neurotransmitter contents and synapse formation may contribute to the pathogenesis of DEE. We show these data in revised Fig. 5, Fig. 9g and Supplementary Fig. 12 and revised the text as follows.

“The cerebellums of *Atp6v0a1^{A512P/A512P}* pups had normal lobules I-X, but showed a reduced molecular layer (ML) (Fig. 4e, f), suggesting that dendrite development of Purkinje cells would be impaired. Moreover, to examine the neuronal connection in wild-type and mutant mice, double immunostaining of the presynaptic marker VGLUT1 and the postsynaptic marker PSD95 was performed in hippocampal CA3 regions and cerebellar lobules. Excitatory synapses are provided by mossy fibers in stratum lucidum and stratum radiatum in the hippocampal regions and by parallel fibers in molecular layers of the cerebellar lobules. VGLUT1 and PSD95 double-positive excitatory synapses were dramatically reduced in both the hippocampal and cerebellar regions in *Atp6v0a1^{A512P/A512P}* pups at P10 (Fig. 5). Thus, *ATP6V0A1^{A512P/A512P}* mice showed defects in neuronal development and synapse formation.” (page 9, paragraph 1)

“We also compared miniature inhibitory postsynaptic currents (mIPSCs) caused by the release of single γ -aminobutyric acid (GABA)-containing vesicles in the same way as mEPSCs and found a significantly larger proportion of small mIPSCs in *Atp6v0a1^{A512P/A512P}* neurons than in *Atp6v0a1^{+/+}* neurons (Supplementary Fig. 12a-f). Thus, the neurotransmitter content of synaptic vesicles was indeed lowered in *Atp6v0a1^{A512P/A512P}* mice, presumably due to the reduced proton pump activity.

Moreover, we also found significant rightward shifts of the distributions of interevent intervals of both mEPSC and mIPSC events in the *Atp6v0a1^{A512P/A512P}* neuron group compared with the *Atp6v0a1^{+/+}* group (Fig. 9g and Supplementary Fig. 12g). This means that both mEPSCs and mIPSCs occurred less frequently in *Atp6v0a1^{A512P/A512P}* neurons. Since miniature postsynaptic events are generated by spontaneous release of synaptic vesicles irrespective of neuronal firing activity, the difference would reflect reduced synapse formation onto *Atp6v0a1^{A512P/A512P}* neurons, as demonstrated for excitatory synapses in the hippocampus and in the cerebellum (Fig. 5).” (page 13)

“We demonstrated that both glutamate and GABA contents of synaptic vesicles was lowered in *Atp6v0a1*^{A512P/A512P} mice (Fig. 9 and Supplementary Fig. 12). To date, pathogenic variants in genes involved in neurotransmission such as *STXBP1*, *SNAP25*, *STX1B*, *DNMI* and *PPP3CA* are known to cause DEE^{23, 24, 25, 26, 27, 28, 29}, suggesting that disturbance of neurotransmission is associated with DEE. Additionally, we demonstrated impaired synaptic connectivity in the hippocampus and cerebellum in *Atp6v0a1*^{A512P/A512P} pups (Fig. 5). A recent report showed that transient expression of ATP6V1A variants identified in DEE patients impaired synapse formation in primary hippocampal neurons, which suggests a novel role for V-ATPase in neuronal development⁷. Our study confirmed the essential role of V-ATPase in synapse formation *in vivo* and suggested that the impaired synaptic connectivity including reduced neurotransmitters and synapse formation may contribute to the pathogenesis of DEE caused by *ATP6V0A1* variants.” (page 15, paragraph 2)

- 3. Functional studies of the v-ATPase activity is needed. Initially the authors examined the effects of ATP6V0a1 variants using the LysoTracker assay. However, the proper way to examine the effects of ATP6V0a1 variants is measurement of v-ATPase activity. Since commercial kits are available, it should be easy to test whether ATP6V0a1 variants really affect the v-ATPase function. LysoTracker is also not the most rigorous index of lysosomal pH and direct ratiometric measurement to precisely determine pH is recommended. A functional readout of the acidification defect using a fluorescent probe to assay proteolytic activity in lysosomes (or an equivalent outcome measure) should be considered.**

Thank you for your instructive comments. To examine V-ATPase activity, we measured lysosomal pH using LysoSensorTM Yellow/Blue dextran (L22460, Invitrogen) in HEK293FT stable cell lines, in which C-terminal 3xHA tagged ATP6V0A1 (ATP6V0A1-3xHA) was expressed. The lysosomal pH values of all ATP6V0A1 mutants were elevated, which indicates that V-ATPase activity in mutants was impaired. We show these data in revised Fig. 3b and revised the text as follows.

“The localizations of three ATP6V0A1 mutants (ATP6V0A1^{A512P}-3xHA, ATP6V0A1^{N534D}-3xHA and ATP6V0A1^{R741Q}-3xHA) were similar to those of wild-type ATP6V0A1 (Fig. 3a). However, the pH measurement of HEK293FT cell lines using LysoSensor Yellow/Blue dextran revealed that lysosomal pH values of all the three types of mutant expressing HEK293FT cells were significantly higher than those of wild-type expressing cells (Fig. 3b).” (page 7, paragraph 2)

- 4. Author need to explain why levels of ATP6V0a1 variants are decreased in Tg mice (Fig 4G). It should be resolved whether or not protein expression is artificially down-regulated and affects net v-ATPase function or whether mutant ATP6V0a1 is more easily degraded. In vivo measurement may not feasible but an in vitro cycloheximide assay will answer this**

question. V0a1 is known to be a glycoprotein. Is there evidence that N741Q, A512P, and N534D mutation also affect glycosylation of the subunit?

Your important comments are greatly appreciated. To examine whether decreased Atp6v0a1^{A512P} protein level in *Atp6v0a1*^{A512P/A512P} mice is caused by impaired protein stability, we performed cycloheximide (CHX) treatment analysis in ATP6V0A1-3xHA stable HEK293FT cell lines. We used the highest-dose condition, 50 μ g/ml CHX for 0-8 h in three independent experiments; however, we could not find any difference among protein levels of wild-type and three mutants during 8-h treatments (Supplementary Fig. 7). These data suggest that the A512P variant could affect protein level *in vivo*, but our *in vitro* test is not sensitive enough to detect difference of protein stability during 8-h periods.

It might be possible that unexpected mutations caused by genome editing would impair *Atp6v0a1* expression. However, we confirmed the 227-bp genomic sequence of *Atp6v0a1*, which covered the entire exon 13 and its flanking 71 and 65-bp intronic sequences spanning the ssODN targeted region; therefore, artificial down-regulation caused by unexpected mutation is less likely, but possible off-target remains elusive.

As we mentioned in reply to comment #1, we examined the glycosylation status of wild-type and three mutants. Three mutant proteins were not affected in glycosylation. We show these data in revised Supplementary Fig. 7c and revised the text as follows.

“These data suggest that *Atp6v0a1*^{A512P} may affect protein stability of Atp6v0a1, which could affect the protein levels of other V-ATPase subunits *in vivo*. To test this possibility, we performed *in vitro* cycloheximide assay using ATP6V0A1-3xHA HEK293FT stable cell lines (Supplementary Fig. 7a, b). ATP6V0A1^{WT}-3xHA proteins were stable and only 20% reduction was observed after 8 h cycloheximide treatment. In this condition, we found no differences among protein levels of wild-type and three mutant ATP6V0A1-3xHA. These data suggest that the A512P variant could affect protein levels *in vivo*, but our *in vitro* test may be not sensitive enough to detect this difference during 8 h periods. ATP6V0A1 proteins undergo glycosylation¹⁵, and glycosylation difference might contribute to the pathogenicity of three variants. However, the glycosylation of three mutants was not changed compared with that of wild-type HA-ATP6V0A1 (Supplementary Fig. 7c).” (page 10, paragraph 1)

“We confirmed the 227-bp genomic sequence of *Atp6v0a1*, which covered the whole of exon 13 and flanking 71 and 65-bp intronic sequences.” (page 20, paragraph 1)”

5. Cross talk between ATP6V0a1 variants and mTOR pathway is not clear. The authors claim that lysosomal dysfunction mediated by ATP6V0a1 variants reduces mTOR signaling. However, it looks like levels of ATP6V0a1 were reduced in P10 cortex (Fig 4G) whereas

levels of phospho-S6 were reduced in P1 cortex (Fig 7C) of the A512P/A512P Tg. Are levels of ATP6V0a1 also reduced in P1 cortex? To support the authors' claim, the deficits are expected to be present by the same early age.

Thank you for your important comments. We examined Atp6v0a1 protein levels in the cortex, hippocampus, and cerebellum at P1. Although the trend for reduced levels in all three regions was recognized in *Atp6v0a1*^{A512P/A512P} pups, the differences did not reach statistical significance (Supplementary Fig. 6d). However, the A512P variant affected V-ATPase activity, as demonstrated by ratiometric measurement of pH (Fig. 3b). Thus, impaired V-ATPase activity may contribute to reduced level of phospho-S6 in P1 cortex (Supplementary Fig. 9). In the revised manuscript, we presented new experimental data at P10 in main figures and presented data from different developmental stages in Supplementary data.

To examine whether the upstream of mTORC1 signaling is altered in mutant brain, we performed immunohistochemistry and immunoblotting with anti-phospho-mTor at serine 2448 (p-mTor, #5536, Cell Signaling) antibody, which is predominantly phosphorylated in mTORC1 signaling (Copp et al., Cancer Res. 2009). In both experiments, the p-mTor level in wild-type and mutant at P10 did not find any statistically significant difference (Fig. 8a-d). Additionally, according to the comments of reviewer #1, we examined nuclear/cytoplasmic localization ratio of TFEB in ATP6V0A1-3xHA HEK293FT stable cell lines to check TFEB activation, but this did not differ among wild-type and three mutants (Supplementary Fig. 10). We show these data in revised Fig. 8a-d and Supplementary Fig. 10 and revised the text as follows.

“V-ATPase regulates mTORC1 signaling activity²¹. We checked expression of phospho-mTor at serine 2448 (p-mTor) and phospho-S6 (pS6) ribosomal protein as markers for mTORC1 signaling activity. Both in immunohistochemistry and immunoblot, p-mTor level was not changed in brain sections between wild-type and respective *Atp6v0a1*^{A512P/A512P} pups (Fig. 8a-d). The pS6 signal in *Atp6v0a1*^{+/+} pups at P10 was strong in pyramidal neurons in both the upper (II/III) and lower (IV/V) layers of the cortex, and in the neurons in CA1 and CA3 regions of the hippocampus (Fig. 8a), and in cerebellar Purkinje cells (Fig. 8b). However, the signal in *Atp6v0a1*^{A512P/A512P} pups was greatly reduced in the cortex and hippocampus (Fig. 8a), and mildly reduced in Purkinje cells (Fig. 8b). Immunoblotting also showed reduced pS6 signals in these three brain regions at P10 (Fig. 8c, e), and in the hippocampus of P1 pups (Supplementary Fig. 9). Moreover, we examined cytoplasmic/nuclear localization of TFEB, mTORC1 signaling regulator, using Flag-TFEB-mClover2 expression vector in ATP6V0A1-3xHA HEK293FT stable cells. We found no differences in localization of TFEB between wild-type and three mutant expressing cells (Supplementary Fig. 10), which suggests that TFEB activation is not altered by ATP6V0A1 variants. Although the upstream signal of mTORC1 signaling and mTORC1-TFEB pathway was not affected by ATP6V0A1 variants, the reduced level

of pS6 in the brain of *Atp6v0a1*^{A512P/A512P} pups suggested impaired mTORC1 activity.” (page 12, paragraph 2)

6. Since the ATP6V0a1 variant was expressed in Ctx/hippo and large dense Cathepsin D accumulation were noticeable in CA3, are there cathepsin D accumulations in Ctx as expected?

To distinguish pre-cathepsin D and mature cathepsin D using immunoblotting, we used new cathepsin D (ab75852, Abcam) antibody and found that cathepsin D activity was decreased (Fig. 6d). This finding was consistent with the fact that both lysosomes and autophagosomes were accumulated in mutant CA3 neurons by electron microscopy (Fig. 7f). With this new cathepsin D antibody, we found cortical neurons with cathepsin D accumulation, but they were fewer than those in hippocampal CA3 and dentate gyrus. We show these data in revised Supplementary Fig. 8 and revised the text as follows.

“However, in *Atp6v0a1*^{A512P/A512P} pups, large dense cathepsin D accumulations were noticeable and they were co-localized with abnormally dispersed distributions of lysosome-associated membrane protein 1 (Lamp1) in the hippocampal CA3 region and dentate gyrus, and they were relatively few in cortex at P10 (Fig. 6c and Supplementary Fig. 8).” (page 11, paragraph 1)

7. Given that A512P/A512P Tg mice died within 2 weeks postnatally, is there any abnormality seen in Hetero-Tg mice at older ages?

We observed *Atp6v0a1*^{A512P/+} mice over 6 months of age, but they showed no abnormal behavior or morphological phenotype. *Atp6v0a1*^{A512P/+} male and female mice also showed natural mating. We revised the text as follows.

“Heterozygous mice of three lines (*Atp6v0a1*^{A512P/+}, *Atp6v0a1*^{R741Q/+} and *Atp6v0a1*^{KO/+}) showed no obvious abnormalities over 6 months of age and were fertile.” (page 8, paragraph 2)

8. Green signal of the IHC is too weak in Fig 5C, so the Lamp2 signal in control mice is difficult to see.

Using another set of antibodies for cathepsin D (ab75852, Abcam) and lysosomal membrane (Lamp1: 1D4B, DSHB), we detected discrete puncta of lysosomes in the wild-type pups (Fig. 6c). Using these antibodies, we confirmed that lysosome membranes were diffusely stained in the cytosol in a subset of CA3 pyramidal neurons in *Atp6v0a1*^{A512P/A512P} brain. Additionally, we examined the lysosomes in hippocampal CA3 neurons of *Atp6v0a1*^{A512P/A512P} pups using electron microscopy. Both

lysosome and autophagosome counts were increased in mutant neurons (Fig. 7f). Thus, we concluded that increased numbers of lysosomes resulted in diffuse staining by Lamp1 and Lamp2 antibodies in mutants (Fig.6 and 7). We show these data in revised Figs. 6c and 7f and Supplemental Fig. 8 and revised the text as follows.

“However, in *Atp6v0a1*^{A512P/A512P} pups, large dense cathepsin D accumulations were noticeable and they were co-localized with abnormally dispersed distributions of lysosome-associated membrane protein 1 (Lamp1) in the hippocampal CA3 region and dentate gyrus, and they were relatively few in cortex at P10 (Fig. 6c and Supplementary Fig. 8).” (page 11, paragraph 1)

“We found no statistically significant difference in LC3-II levels in the hippocampus of *Atp6v0a1*^{A512P/A512P} pups using immunoblotting; however, transmission electron microscopy showed that excessive autophagosomes with double membrane (Fig. 7f, blue), lysosomes with single membrane (Fig. 7f, brown) and rough endoplasmic reticulum were present in mutant CA3 neurons, but not in cerebellum.” (pages 11 to 12)

9. Please state how many mice are used for each IHC study and document that the lysosomal abnormality is consistently robust in all Tg mice.

Thank you for your important advice. We analyzed three or more independent mice for each genotype for IHC study and analyzed two independent samples of each genotype for electron microscopy analysis. We added clear descriptions in Subjects and Methods. We consistently observed the lysosomal and autophagosome abnormalities as shown by cathepsin D, Lamp1/2 and LC3 immunofluorescence and electron microscopy in all Tg mice with variable severity. Thus, we quantified the lysosomal and autophagosome abnormalities using immunoblotting (Figs. 6 and 7), and found statistical differences in some brain regions.

REVIEWERS' COMMENTS

Reviewer #1 (Remarks to the Author):

The authors have provided comprehensive and appropriate responses to the reviews of the previous version of the manuscript. I have no further comments and I recommend publication of the study.

Reviewer #2 (Remarks to the Author):

For this revised version of their report, the authors have provided an extensive amount of additional high quality data in support of their conclusions. They have fully addressed the concerns raised in the original review. The report is a significant contribution to understanding vATPase function and relevance to vATPase deficiency disorders.

Response to reviewers:

Reviewer #1 (Remarks to the Author):

- 1. The authors have provided comprehensive and appropriate responses to the reviews of the previous version of the manuscript. I have no further comments and I recommend publication of the study.**

Thank you for your positive assessments.

Reviewer #2 (Remarks to the Author):

- 1. For this revised version of their report, the authors have provided an extensive amount of additional high quality data in support of their conclusions. They have fully addressed the concerns raised in the original review. The report is a significant contribution to understanding vATPase function and relevance to vATPase deficiency disorders.**

Thank you for your positive assessments.